# Visual Representation Learning for World Models by Predicting Fine-Grained Motion

## Abstract

Originating from model-based reinforcement learning (MBRL) methods, algorithms based on world models have been widely applied to boost sample efficiency in visual environments. However, existing world models often struggle with irrelevant background information and omit moving tiny objects that can be essential to tasks. To solve this problem, we introduce the Motion-Aware World Model (MAWM), which incorporates a fine-grained motion predictor and entails action-conditional video prediction with a motion-aware mechanism. The mechanism yields compact and robust representations of environments, filters out extraneous backgrounds, and keeps track of the pixel-level motion of objects. Moreover, we demonstrate that a world model with action-conditional video prediction can be interpreted as a variational autoencoder (VAE) for the whole video. Experiments on the Atari 100k benchmark show that the proposed MAWM outperforms current prevailing MBRL methods. We further show its state-of-the-art performance across challenging tasks from the DeepMind Control Suite.

## 1 Introduction

Recent model-based reinforcement learning (MBRL) algorithms utilize world models (Kalweit & Boedecker, 2017) to capture the dynamics of the environment and endow agents with the ability to learn compact representations from high-dimensional images (Watter et al., 2015; Ebert et al., 2018; Hafner et al., 2019b; Zhang et al., 2019), imagine future frames (Denton et al., 2017; Hafner et al., 2019a; Kaiser et al., 2020) and plan (Chua et al., 2018; Schrittwieser et al., 2020; Ye et al., 2021; Wang et al., 2024). As a notable example of MBRL approaches, DreamerV3 (Hafner et al., 2023) learns a world model, which consists of a recurrent state-space model (RSSM; Hafner et al., 2019b), a variational autoencoder (VAE; Kingma, 2013), and predictors for accessible signals. Then an actor-critic network utilizes predictions from the world model to learn long-horizon behaviors.

Due to aleatoric uncertainty and epistemic uncertainty (Lakshminarayanan et al., 2017), it is difficult for world models to have a perfect prediction for rewards. Prediction errors often hinder a guarantee of policy improvement for a model-based method (Janner et al., 2019). When it comes to visually complex environments with many moving small objects, the situation gets even worse. Motivated by diffusion models (Song et al., 2021; Karras et al., 2022; Ho et al., 2022c), Alonso et al. (2024) designed a diffusion world model which predicts future frames conditioning on past observations and actions to keep small details in the visual inputs. To avoid reconstruction of irrelevant details such as textures or environment noise at the expense of smaller but important elements, Sun et al. (2024) randomly masked a portion of pixels in the video clip to reduce the spatio-temporal redundancy. However, the methods proposed above failed to deal with moving tiny objects and neglected their connections with tasks.

Current representation learning methods in MBRL via the task of image reconstruction could not concentrate on the moving object that indicates the result of actions but may lay much emphasis on the background, which occupies most of the area of images. To give an illustration, imagine a moving tiny object in an environment, a neural network model that simply reconstructs the images of the environment can exhibit low error enough. That is to say, the model is not encouraged to focus on the tiny object but pays attention to the background. Representation learning via video prediction may tackle the above problem. However, there are often subtle differences between neighboring

frames. It is necessary to develop an appropriate motion-aware mechanism to address the above problem.

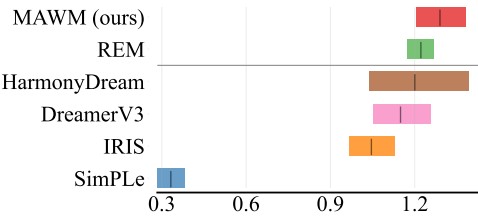

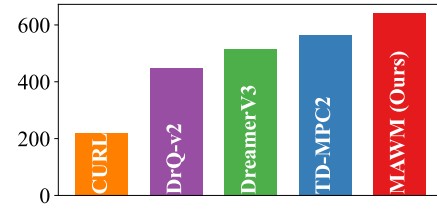

(a) Human-normalized score mean on Atari 100k    (b) Average score on DeepMind Control Suite

Figure 1: Comparison of methods on Atari 100k and challenging tasks from DeepMind Control Suite benchmarks. MAWM achieves consistently strong performance with fixed parameters for all tasks across both domains.

To draw the attention of the agent to moving tiny objects and inter-frame discrepancy, we present MAWM (Motion-Aware World Model), a deep neural network framework that learns compact world models via fine-grained motion prediction and action-conditional video prediction. MAWM focuses on moving objects in video and pays attention to meaningful small objects via pixel-level attention mechanisms. MAWM has an adaptive control scheduler to deal with rapid changes in the environment, similar to the causes of visually-induced dizziness in humans, which enables robust representation learning for the foreground region in different environments. We conduct experiments and demonstrate the strong adaptability of MAWM for diverse control scenarios. The main contributions of this work are summarized as follows.

- We design a framework of world models, called MAWM, which incorporates a new motion-aware mechanism and learns visual representations via video prediction and motion prediction with an Adaptive Motion-Aware Scheduler (AMAS).
- We introduce a novel theoretical model named Recurrent State Space Model for Video Prediction (RSSM-VP), which establishes the foundation for applying RSSM to world model learning via video prediction, and infer the training objective of MAWM from it.
- We show MAWM masters visual control tasks across diverse domains, encompassing discrete and continuous actions. Specifically, MAWM outperforms DreamerV3 by a large margin, on both Atari 100k and challenging tasks from DeepMind Control Suite.

## 2 RELATED WORK

### 2.1 MODEL-BASED REINFORCEMENT LEARNING

Recent years have witnessed the growing importance of sample-efficient reinforcement learning in complex visual environments (Hafner, 2022) and MBRL has been a research focus in recent decades (Sutton, 1991; Moerland et al., 2023). Currently, MBRL reduces the number of interactions between the agent and the environment by learning policy within a world model. Ha & Schmidhuber (2018) first proposed a simple world model composed of Mixture Density Network (Graves, 2013) combined with an LSTM (Hochreiter, 1997) model and a VAE (Kingma, 2013) model to learn the dynamics in visual environments. Dreamer, a notable series of methods (Hafner et al., 2019a; 2020; 2023), is based on the recurrent state-space model, which enables forward predictions purely in latent space. Descendants of RSSM, such as C-RSSM (Gumbsch et al., 2023) and HRSSM (Sun et al., 2024), were proposed to learn hierarchical and robust latent representations. However, RSSM and its variants were limited to representation learning via image reconstruction (Ha et al., 2023). In contrast, RSSM-VP is a universal theoretical model that enables world models based on RSSM to learn from video prediction and is applicable to other variants of RSSM.

Encouraged by the huge success of Transformer architecture (Vaswani, 2017) in natural language processing and computer vision, several works attempted to use a transformer-based world model to

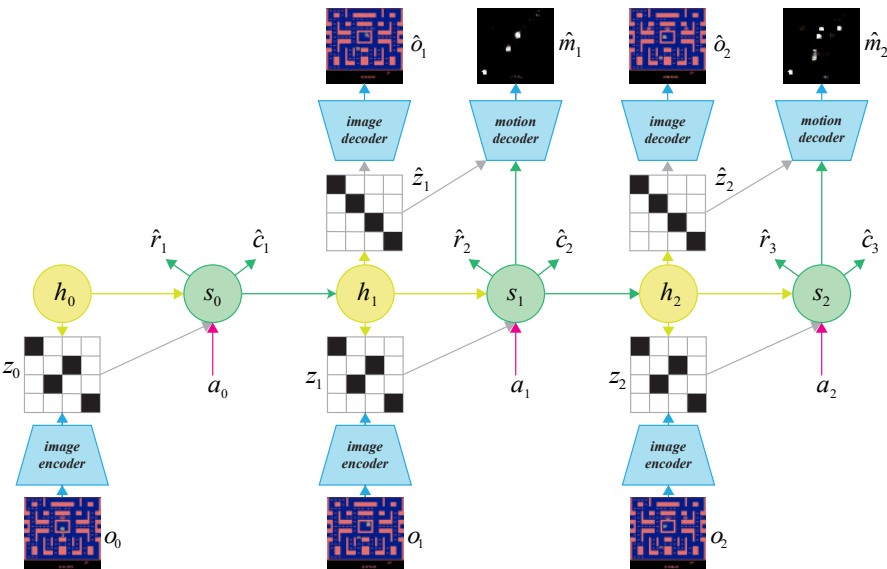

Figure 2: MAWM architecture that predicts future frames conditioned on the history of frames $o_{0:t-1}$ and actions $a_{0:t-1}$. The image decoder predicts the next-frame image $\hat{o}_t$ via deterministic states $h_t$ and prior stochastic states $\hat{z}_t$. The representation model combines features extracted by the image encoder from frames $o_t$ with deterministic states $h_t$ to obtain posterior stochastic states $z_t$. The motion decoder learns to predict masks for the motion of objects by minimizing the focal loss (Lin et al., 2017) for binary classification.

learn the dynamics in environments, such as Transdreamer (Chen et al., 2022), IRIS (Micheli et al., 2023), MWM (Seo et al., 2023), TWM (Robine et al., 2023), STORM (Zhang et al., 2024) and REM (Cohen et al., 2024). Planning using the world model at inference time can improve the accuracy of action selections. Building upon MuZero (Schrittwieser et al., 2020) which leveraged Monte Carlo Tree Search (Coulom, 2006), EfficientZero (Ye et al., 2021) introduced a self-supervised consistency loss and used imagined rollouts with current policy to obtain the value target. By utilizing sampled-based Gumble search (Danihelka et al., 2022) and search-based value estimation, EfficientZero V2 (Wang et al., 2024) has been the state-of-the-art algorithm on the Atari 100k benchmark so far. Meanwhile, some works (Seo et al., 2022; Wu et al., 2024) tried to pre-train a model from off-the-shelf video via unsupervised representation learning and stack an action-conditional latent prediction model on top of the pre-trained model. TD-MPC2 (Hansen et al., 2024) optimizes local trajectories in a learned implicit world model without the decoder. HarmonyDream (Ma et al., 2024a) proposed task harmonizers, *i.e.*, learnable parameters, with which world models can balance various loss terms automatically. To our best knowledge, MAWM is the first world model that incorporates a motion-aware mechanism, which can be incorporated into existing MBRL methods to capture moving tiny objects.

## 2.2 ACTION-CONDITIONAL VIDEO PREDICTION

Oh et al. (2015) made and evaluated long-term predictions on visual images conditioned on actions in Atari games in their pioneering research. They extracted high-level feature vectors from a fixed number of frames using convolutional neural networks and utilized LSTM to capture temporal correlations among these feature vectors. Later works further improved this architecture by adding skip connections between the convolutional encoder and decoder (Finn et al., 2016), discretizing feature vectors, and utilizing a variational autoencoder to get a stochastic model (Kaiser et al., 2020). COSMOS (Sehgal et al., 2024) extracted objects in the images and applied the neurosymbolic attention mechanism that binds these objects to learned rules of interaction from an object-centric perspective. In recent years, the focus has shifted towards generative video prediction, which makes it necessary to have a profound understanding of the physical principles (Ming et al., 2024a). Alonso et al. (2024) proposed a diffusion world model conditioned on a sequence of images and actions. To estimate and

generate the next observation, they concatenate the past images to a noisy image channel-wise and input actions through adaptive group normalization layers (Zheng et al., 2020). However, they simply used predicted frames as states for policy learning. By contrast, our algorithm utilizes the latent states for policy learning on the foundation of our proposed RSSM-VP.

## 3 METHODS

Visual reinforcement learning is formalized as a Partially Observable Markov Decision Process (POMDP; Kaelbling et al., 1998) with image observations $o_t \in \Omega \subseteq \mathbb{R}^{h \times w \times 3}$, actions $a_t \in \mathcal{A}$, rewards $r_t \in \mathbb{R}$, states $s_t$, and a discount factor $\gamma \in (0, 1]$. An agent takes an action according to the policy $\pi(\cdot|o_{\leq t}, a_{<t})$, which is a mapping from the history of past observations and actions to a probability distribution on actions to take. The object is to learn a policy $\pi$ that maximizes the expected value of accumulated discounted reward $\mathbb{E}_\pi[\sum_{t=0}^{\infty} \gamma^t r_t | s_0 = s]$.

We focus on visual representation learning for world models. We first provide the framework of MAWM and how to learn latent representations and dynamics for our world model in Section 3.1. We then present details of visual representation learning by fine-grained motion prediction and action-conditional video prediction in Section 3.2 and 3.3, respectively. We summarize the training protocol of MAWM in Appendix C.

### 3.1 MAWM FRAMEWORK

**Components** We utilize images $o_t$, rewards $r_t$, motion hints $m_t$, and episode continuation flags $c_t$ to learn the world model in a self-supervised manner. MAWM consists of the following components:

$$
\begin{aligned}
\text{Sequence model:} \quad & h_t = h_\phi(s_{t-1}, a_{t-1}) \\
\text{Representation model:} \quad & z_t \sim q_\phi(z_t|h_t, o_t) \\
\text{Dynamics model:} \quad & \hat{z}_t \sim p_\phi(\hat{z}_t|h_t) \\
\text{Video predictor:} \quad & \hat{o}_t \sim p_\phi(\hat{o}_t|h_t, \hat{z}_t) \\
\text{Motion predictor:} \quad & \hat{m}_t \sim p_\phi(\hat{m}_t|s_t, \hat{z}_t) \\
\text{Reward predictor:} \quad & \hat{r}_t \sim p_\phi(\hat{r}_t|s_t) \\
\text{Continue predictor:} \quad & \hat{c}_t \sim p_\phi(\hat{c}_t|s_t),
\end{aligned}
\tag{1}
$$

where $s_t$ is the hidden state, $z_t$ the posterior stochastic state, and $h_t$ the deterministic state. Though $s_t$ can be a function of $z_t$ and $h_t$ theoretically, we concatenate $h_t$ to $z_t$ into the hidden state $s_t$ in practice. The stochastic state $z_t$ is sampled from a vector of categorical distributions and the prior stochastic state $\hat{z}_t$ is sampled similarly. Detailed architecture of each component is presented in Appendix B.

**Loss function** Given a sequence of images $o_{0:T}$, motion hints $m_{1:T}$, actions $a_{0:T-1}$, rewards $r_{0:T-1}$, continuation flags $c_{0:T-1}$, parameters $\phi$ of world model are optimized end-to-end to minimize the following loss

$$
\mathcal{L}(\phi, \sigma) = \sum_{t=1}^{T} \mathbb{E}_{q(s_{t-1}|o_{<t}, a_{<t-1})} \Big[ \sum_{x \in \{m,o,r,c,dyn,rep\}} \beta_x (\sigma_x \mathcal{L}_t^x(\phi) + \log(1 + \sigma_x)) \Big],
\tag{2}
$$

where $\beta_x$ are the weights of loss terms and $\sigma_x$ are learnable parameters, dubbed as harmonizers (Ma et al., 2024a), which rescale losses during training. Reward loss $\mathcal{L}_t^r(\phi)$ and continuation loss $\mathcal{L}_t^c(\phi)$ are both negative log-likelihood losses. By contrast, details of motion loss $\mathcal{L}_t^m(\phi)$ and video prediction loss $\mathcal{L}_t^o(\phi)$ are demonstrated in Section 3.2 and 3.3, respectively. Dynamics loss $\mathcal{L}_t^{dyn}(\phi)$ and representation loss $\mathcal{L}_t^{rep}(\phi)$ de facto constitute the KL loss $D_{KL}(q(s_t|o_{\leq t}, a_{<t})||p(s_t|s_{t-1}, a_{t-1}))$ via KL balancing (Hafner et al., 2020), differing in the domain of stop-gradient operator $\text{sg}(\cdot)$ and their loss scale. To avoid a trivial solution where the prior stochastic state $\hat{z}_t$ contains not enough information about images, free bits (Kingma et al., 2016) clipping the dynamics and representation losses are employed:

$$
\begin{aligned}
\mathcal{L}_t^{dyn}(\phi) &= \max\left(1, D_{KL}\left[\text{sg}(q_\phi(z_t|s_t))||p_\phi(z_t|h_t)\right]\right) \\
\mathcal{L}_t^{rep}(\phi) &= \max\left(1, D_{KL}\left[q_\phi(z_t|s_t)||\text{sg}(p_\phi(z_t|h_t))\right]\right).
\end{aligned}
\tag{3}
$$

**Behavior learning** To learn behaviors from imagined hidden states within world models, we opt for the standard actor-critic framework from DreamerV3 (Hafner et al., 2023). It is noteworthy that the prediction of motion hints or frames is unnecessary during policy learning and thus computational overhead of the video predictor and the motion predictor can be avoided.

## 3.2 FINE-GRAINED MOTION PREDICTION

As we aim to learn motion-aware representation by explicitly predicting fine-grained motion, the motion map for every frame is necessary. To avoid labeling motion information by hand, we first use an adaptive Gaussian Mixture Model (GMM) for pixel-level motion extraction, which involves judgment of whether the pixel belongs to background or not (Zivkovic, 2004; Zivkovic & Van Der Heijden, 2006) and outputs binary masks $m_t \in \mathbb{R}^{h \times w}$. The important components related to the proposed motion-aware mechanism are elaborated below.

**Image encoder** Attention plays an essential role in human perception by selective focus on interesting parts of the environment, especially motions and moving objects. It has been proposed that bottom-up sensory-driven mechanisms are parts of mechanisms of human attention (Ungerleider & G., 2000; Petersen & Posner, 2012). To focus on important features and the regions of interest, we integrate into our image encoder network the Convolutional Block Attention Module (CBAM; Woo et al., 2018), as detailed in Appendix B.1.

**Motion predictor** To entail the world model to learn motion-aware representations, we design a motion predictor to capture moving objects and changes in the environment. We use a decoder network to extract motion hints $\hat{m}_t$ from video and estimate $p_{i,j} \in [0, 1]$, probability of the foreground class of every pixel in the image, as shown in Equation 1. The total loss for motion prediction is the sum of focal loss (Lin et al., 2017) of every pixel in the image

$$\sum_{i=0}^{h-1} \sum_{j=0}^{w-1} FocalLoss(p_{i,j}^t) = \sum_{i=0}^{h-1} \sum_{j=0}^{w-1} -\alpha(1-p_{i,j}^t)^\gamma \log p_{i,j}^t, \tag{4}$$

where $\alpha \in [0, 1]$ balances the importance of foreground and background loss. The larger $\alpha$ is, the more emphasis a world model puts on the foreground. $\gamma \geq 0$ is a parameter to deal with pixels that are hard to classify. The auxiliary variable $p_{i,j}^t$ is equal to $p_{i,j}$ when the binary mask of a pixel is 1. Otherwise, $p_{i,j}^t = 0$.

**Adaptive motion-aware scheduler** When the environment changes rapidly, the background will take over from motion clues in the binary masks predicted by background subtraction methods. To address the above issue, we develop an adaptive motion-aware scheduler (AMAS) that can automatically terminate the focus on motion hints, just as humans feel dizzy when confronted with complex patterns or movement (Kim et al., 2020; Keshavarz et al., 2023). Give the threshold of the number of pixels that agents can pay attention to, denoted as $r_{\text{dizzy}}$, AMAS is a function that depends on motion masks $m_t$:

$$AMAS(m_t) = \mathbb{I}(\sum_{i=0}^{h-1} \sum_{j=0}^{w-1} m_{t,i,j} > r_{\text{dizzy}} \times h \times w), \tag{5}$$

where $\mathbb{I}$ is the indicator function. From Equation 4, the motion prediction loss with AMAS is:

$$\mathcal{L}_t^{\text{m}}(\phi) = -AMAS(m_t) \sum_{i=0}^{h-1} \sum_{j=0}^{w-1} \alpha(1-p_{i,j}^t)^\gamma \log p_{i,j}^t \tag{6}$$

## 3.3 ACTION-CONDITIONAL VIDEO PREDICTION

Ma et al. (2024b) formulates Action-conditional World Model (AWM) as $\hat{s}_{t+1} = g(s_0, a_0, ..., a_t)$ and demonstrates that actions are sufficient to predict future states in stochastic environments just as they are sufficient in a deterministic Markov Decision Process. Since an agent with only visual inputs may not be able to capture the actual hidden state in a POMDP, it's unrealistic for the agent to predict frames only with an action sequence in the far future when it comes to a stochastic environment. Nevertheless, we hypothesize that agents can exactly predict the next frame given the history of observations and actions. In contrast to the vanilla RSSM (Hafner et al., 2019b) designed for image

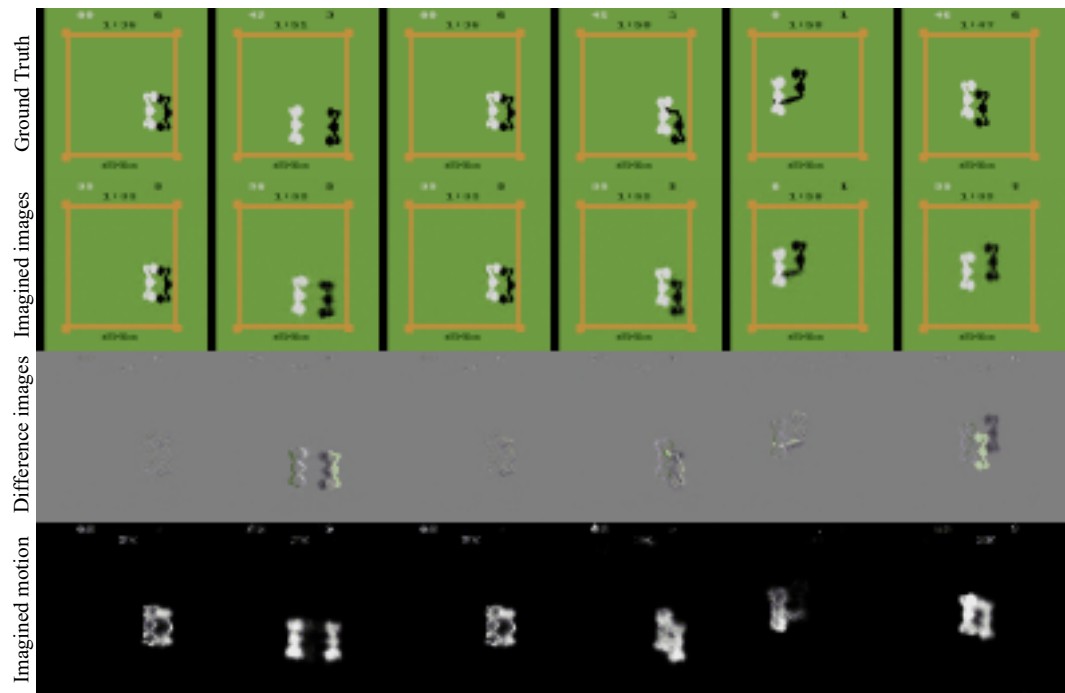

Figure 3: Imagined images and motion when the imagination step equals 9. The third row is the difference between the ground truth and imagined images. Each column is sampled from a different trajectory. It is noteworthy that positions of motion estimated by the motion predictor remain accurate even when imagined images are different in positions of objects from the ground truth, which are particularly noticeable in the second and the last column.

reconstruction, we employ RSSM to model the stochasticity of future frames. That is to say, we predict the next frame without access to it, which can be regarded as an action-conditional video prediction problem. Furthermore, the exploitation of RSSM for video prediction can provide latent states for policy learning while other MBRL methods based on video prediction only provide image embeddings (Micheli et al., 2023; Cohen et al., 2024) for policy learning.

We here present RSSM-VP, a theoretical model that changes the traditional concept of RSSM and makes it applicable to video prediction. Instead of traditionally interpreting the world model as a sequential VAE, we interpret the world model as a single VAE for video and demonstrate that the VAE for video can be decomposed into the representation model and the dynamics model. Furthermore, we derive the training objective from the VAE for video for completeness and clarity. Finally, we propose motion-aware video prediction loss, in concert with motion prediction loss in Section 3.2, to learn compact motion-aware representations.

**VAE for video** Given the first frame and a sequence of actions, we interpret the world model as a VAE for the whole video, where a video encoder $q_\phi(s_{0:T-1}|a_{0:T-1}, o_{1:T}, o_0) = \prod_{t=0}^{T-1} q_\phi(s_t|o_{\leq t}, a_{<t})$ parameterizes the approximate posterior distribution of all hidden states from the video, and a state transition function $p_\phi(s_{0:T-1}|a_{0:T-1}, o_0)$ parameterizes the prior distribution of hidden states without the video input, which can be regarded as the video decoder. We formulate the latent dynamics model as $p_\phi(o_{1:T}, s_{0:T-1}|a_{0:T-1}, o_0) = \prod_{t=1}^{T} p_\phi(o_t|s_{t-1}, a_{t-1}, o_0) p_\phi(s_{t-1}|s_{t-2}, a_{t-2}, o_0))$, where $p_\phi(s_0|s_{-1}, a_{-1}, o_0)$ is defined as $p_\phi(s_0|o_0)$ and $p_\phi(s_{t-1}|s_{t-2}, a_{t-2}, o_0))$ is the state transition function conditioned on the first frame for time step $t$ ($1 < t \leq T$). As illustrated in Appendix A.1, we can decompose the video encoder into the representation model $q_\phi(z_t|h_t, o_t)$ and the state transition function into the dynamics model $p_\phi(\hat{z}_t|h_t)$ for every time step $t$. Thus the video prediction problem for world models with RSSM can be regarded as a problem of next-frame prediction in sequence.

**Training objective** Though Hafner et al. (2019b) has proved an objective for world model training via image reconstruction, it is unclear what the training objective is to learn a world model with RSSM for video prediction. A direct objective is to maximize the log-likelihood of video data. Therefore, we derive the Evidence Lower BOund (ELBO) of the log-likelihood conditioned on the action sequence and the first frame. We only describe the video prediction loss here and omit the symbol $\phi$ for simplicity. Using importance weighting and Jensen's inequality, as shown in Appendix A.2, we can obtain the ELBO as follows:

$$
\ln p(o_{1:T}|a_{0:T-1}, o_0) \triangleq \ln \mathbb{E}_{p(s_{0:T-1}|a_{0:T-1}, o_0)} \left[ \prod_{t=1}^{T} p(o_t|s_{t-1}, a_{t-1}) \right]
$$

$$
\geq \sum_{t=1}^{T} \mathbb{E}_{q(s_{t-1}|o_{<t}, a_{<t-1})} \left[ \ln p(o_t|s_{t-1}, a_{t-1}) - D_{\mathrm{KL}}\left(q(s_t|o_{\leq t}, a_{<t})||p(s_t|s_{t-1}, a_{t-1})\right) \right] \tag{7}
$$

$$
- D_{\mathrm{KL}}(q(s_0|o_0)||p(s_0|o_0)),
$$

where $D_{\mathrm{KL}}(\cdot||\cdot)$ is the Kullback-Leibler divergence of two distributions and $q(s_0|o_{\leq 0}, a_{<0})$ is defined as $q(s_0|o_0)$. We can set $q(s_0|o_0) \equiv p(s_0|o_0)$ to save the hassle of dealing with inputs with different dimensions.

**Motion-aware video prediction loss** To entail more concentration on the area where changes occur, we propose the motion-aware video prediction loss instead of the log-likelihood loss, which is also a part of our motion-aware mechanism. Specifically, given the ground truth $o_t$ and outputs of the video predictor $\hat{o}_t$, with the AMAS from Equation 5, the video prediction loss is:

$$
\mathcal{L}_t^o(\phi) = e_t + \omega AMAS(m_t)(m_t - 1) \odot e_t, \tag{8}
$$

where $e_t = (o_t - \hat{o}_t) \odot (o_t - \hat{o}_t)$, $\omega \in [0, 1]$ is the motion-aware weight to balance attention of the whole images and attention of motion hints. If every pixel of masks $m_t$ equals 1 or the AMAS is disabled, then video prediction loss $\mathcal{L}_t^o(\phi)$ is equivalent to the mean squared error between predicted video frames and the ground truth.

## 4 EXPERIMENTS

We evaluate our world model MAWM on the well-established Atari 100k Benchmark for data efficiency. To further explore the ability of MAWM, we also conducted experiments on DeepMind Control Suite (Tassa et al., 2018). Details for benchmarks and baselines are included in Section 4.1. A comprehensive evaluation of results on the two benchmarks is presented in Section 4.2. Ablation studies of the key elements proposed for MAWM are shown in Section 4.3. We also include an additional experiment on the DMC-GB2 (Almuzairee et al., 2024) benchmark in Appendix L to evaluate the generalization ability of MAWM.

### 4.1 EXPERIMENTAL SETUP

**Atari 100k benchmark** is comprised of 26 different Atari video games (Bellemare et al., 2013) across a diverse range of genres. The benchmark challenges general algorithms to sample-efficient learning within 100k interactions in various environments, equivalent to 400k environment steps with 4 repeated actions or 2 hours of human gameplay. Standard measurement for a game is human-normalized score (HNS; Mnih et al., 2015), calculated as $HNS = \frac{s_a - s_h}{s_h - s_r}$, where $s_a$ denotes the game score of the algorithm, $s_h$ denotes the game score of a human player, and $s_r$ denotes the game score of a random policy.

**DeepMind Control Suite** is a set of classical continuous control tasks for robotics and reinforcement learning research. On this benchmark, we restrict inputs of algorithms to high-dimensional images. By convention (Hafner et al., 2023), the number of environment steps is 1M, which amounts to 500k interactions with 2 repeated actions. We select hard tasks (Hubert et al., 2021) that are not satisfactorily resolved by existing MBRL methods, resulting in 8 tasks, which are listed in Table 2.

We choose competent baselines for both domains. On the Atari 100k benchmark, besides DreamerV3 (Hafner et al., 2023) and HarmonyDream (Ma et al., 2024a), we choose world models via

Table 1: Game scores and human normalized aggregate metrics on the 26 games of the Atari 100k benchmark. We highlight the highest and the second highest scores among all baselines in bold and with underscores, respectively.

| Game | Random | Human | SimPLe | IRIS | DreamerV3 | HarmonyDream | REM | MAWM (Ours) |
|---|---|---|---|---|---|---|---|---|
| Alien | 227.8 | 7127.7 | 616.9 | 420.0 | _1024.9_ | **1179.3** | 607.2 | 776.4 |
| Amidar | 5.8 | 1719.5 | 88.0 | 143.0 | 130.8 | **166.3** | 95.3 | _144.2_ |
| Assault | 222.4 | 742.0 | 527.2 | _1524.4_ | 723.6 | 701.7 | **1764.2** | 883.4 |
| Asterix | 210.0 | 8503.3 | 1128.3 | 853.6 | 1024.2 | _1260.2_ | **1637.5** | 1096.9 |
| BankHeist | 14.2 | 753.1 | 34.2 | 53.1 | **1018.9** | 627.1 | 19.2 | _742.6_ |
| BattleZone | 2360.0 | 37187.5 | 5184.4 | _13074_ | 11246.7 | 11563.3 | 11826 | **13372.0** |
| Boxing | 0.1 | 12.1 | 9.1 | 70.1 | 84.8 | _86.0_ | **87.5** | 85.4 |
| Breakout | 1.7 | 30.5 | 16.4 | _83.7_ | 26.9 | 34.9 | **90.7** | 71.8 |
| ChopperCommand | 811.0 | 7387.8 | 1246.9 | _1565.0_ | 709.7 | 627.0 | **2561.2** | 904.0 |
| CrazyClimber | 10780.5 | 35829.4 | 62583.6 | 59324.2 | _81414.7_ | 54687.3 | 76547.6 | **89038.6** |
| DemonAttack | 152.1 | 1971.0 | 208.1 | _2034.4_ | 226.5 | 267.0 | **5738.6** | 152.2 |
| Freeway | 0.0 | 29.6 | 20.3 | _31.1_ | 9.5 | 0.0 | **32.3** | 0.0 |
| Frostbite | 65.2 | 4334.7 | 254.7 | 259.1 | 251.7 | **1937.9** | 240.5 | _692.6_ |
| Gopher | 257.6 | 2412.5 | 771.0 | 2236.1 | 4074.9 | **9564.7** | _5452.4_ | 4415.8 |
| Hero | 1027.0 | 30826.4 | 2656.6 | 7037.4 | 4650.9 | **9865.3** | 6484.8 | _8801.8_ |
| Jamesbond | 29.0 | 302.8 | 125.3 | **462.7** | 331.8 | 327.8 | _391.2_ | 337.2 |
| Kangaroo | 52.0 | 3035.0 | 323.1 | 838.2 | 3851.7 | **5237.3** | 467.6 | _3875.6_ |
| Krull | 1598.0 | 2665.5 | 4539.9 | 6616.4 | _7796.6_ | 7784.0 | 4017.7 | **8729.6** |
| KungFuMaster | 258.5 | 22736.3 | 17257.2 | 21759.8 | 18917.1 | 22131.7 | **25172.2** | _23434.6_ |
| MsPacman | 307.3 | 6951.6 | 1480.0 | 999.1 | _1813.3_ | **2663.3** | 962.5 | 1580.7 |
| Pong | -20.7 | 14.6 | 12.8 | 14.6 | 17.1 | _20.0_ | 18.0 | **20.1** |
| PrivateEye | 24.9 | 69571.3 | 58.3 | **100.0** | 47.4 | -198.6 | _99.6_ | -472.5 |
| Qbert | 163.9 | 13455.0 | 1288.8 | 745.7 | 873.2 | **1863.3** | 743 | _1664.4_ |
| RoadRunner | 11.5 | 7845.0 | 5640.6 | 9614.6 | **14478.3** | 12478.3 | _14060.2_ | 12518.6 |
| Seaquest | 68.4 | 42054.7 | _683.3_ | 661.3 | 479.1 | 540.7 | **1036.7** | 557.9 |
| UpNDown | 533.4 | 11693.2 | 3350.3 | 3546.2 | _20183.2_ | 10007.1 | 3757.6 | **28408.2** |
| #Superhuman(↑) | 0 | N/A | 1 | 10 | 10 | 9 | **12** | **12** |
| Mean(↑) | 0.0 | 1.000 | 0.332 | 1.046 | 1.150 | 1.200 | _1.222_ | **1.290** |
| Median(↑) | 0.0 | 1.000 | 0.134 | 0.289 | 0.575 | _0.634_ | 0.280 | **0.651** |

Table 2: Scores achieved across eight challenging tasks from DeepMind Control Suite with a budget of 500k interactions. We highlight the highest and the second highest scores among all baselines in bold and with underscores, respectively.

| Task | CURL | DrQ-v2 | DreamerV3 | TD-MPC2 | MAWM (Ours) |
|---|---|---|---|---|---|
| Acrobot Swingup | 5.1 | 128.4 | 210.0 | 295.3 | **452.1** |
| Cartpole Swingup Sparse | 236.2 | 706.9 | **792.9** | _790.0_ | 666.7 |
| Cheetah Run | 474.3 | 691.0 | _728.7_ | 537.3 | **874.3** |
| Finger Turn Hard | 215.6 | 220.0 | 810.8 | _885.2_ | **935.0** |
| Hopper Hop | 152.5 | 189.9 | **369.6** | 302.9 | _311.5_ |
| Quadruped Run | 141.5 | _407.0_ | 352.3 | 283.1 | **648.7** |
| Quadruped Walk | 123.7 | **660.3** | 352.6 | 323.5 | _580.3_ |
| Reacher hard | 400.2 | 572.9 | 499.2 | **909.6** | _654.9_ |
| Mean(↑) | 218.6 | 447.1 | 514.5 | _540.9_ | **640.4** |
| Median(↑) | 184.1 | _490.0_ | 434.4 | 430.4 | **651.8** |

video prediction, including SimPLe (Kaiser et al., 2020), IRIS (Micheli et al., 2023), and REM (Cohen et al., 2024). Apart from DreamerV3 and TD-MPC2 (Hansen et al., 2024) , our baselines also include CURL (Laskin et al., 2020) and DrQ-v2 (Yarats et al., 2022), which are model-free RL methods. As suggested by aforementioned methods (Micheli et al., 2023; Robine et al., 2023; Cohen et al., 2024; Zhang et al., 2024), we here exclude lookahead search methods because we aim to learn a compact and meaningful world model itself. Nevertheless, lookahead search techniques like Monte-Carlo Tree Search (Coulom, 2006) and Gumbel search (Danihelka et al., 2022) can be integrated with MAWM at the expense of computational burden. Appendix J provides a broader comparison to lookahead search methods.

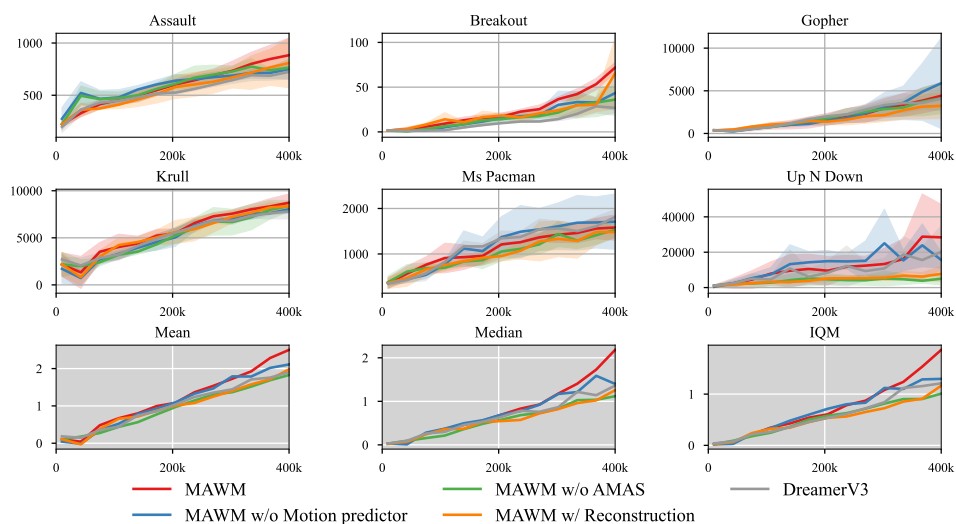

Figure 4: Ablation studies of key contributions of MAWM on Atari 100k. The shaded region indicates the standard deviation.

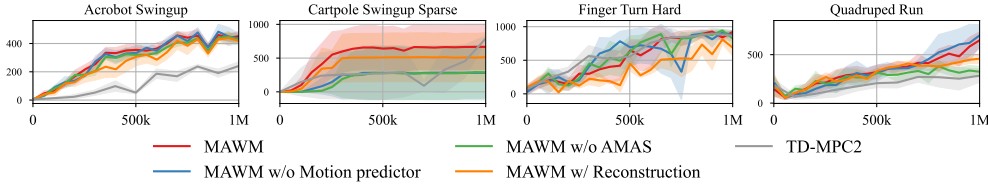

Figure 5: Ablation studies key contributions of MAWM on DeepMind Control Suite.

## 4.2 PERFORMANCE EVALUATION

**Atari 100k benchmark** The score of each game and aggregate performance metrics on the Atrai 100k benchmark are showcased in Table 1. MAWN was trained from scratch and evaluated by conducting 100 evaluation episodes at the end of training. The results for Random, Human, Sim-PLe, and REM are sourced from previous work (Cohen et al., 2024). We reproduce the results of DreamerV3 and HarmonyDream and implementation details of both algorithms can be found in Appendix D.2 and D.3. MAWM obtains a mean human-normalized score of 129.0%, surpassing all the baselines. Following the recommendations of (Agarwal et al., 2021) on the reliable evaluation for reinforcement learning methods, we also report stratified bootstrap confidence intervals for all aggregate metrics in Appendix E.

**DeepMind Control Suite** Table 2 displays the scores across challenging tasks from DeepMind Control Suite. MAWM reaches a mean score of 640.6 and a median score of 651.8, setting new state-of-the-art results for RL methods. MAWM outperforms all baselines on four out of eight challenging tasks and performs consistently well on the remaining tasks, except Cartple Swingup Sparse. Due to a sparse reward setting in this task, the agent may never obtain any positive feedback from the environment under some seeds, which strangles policy learning within the world model.

## 4.3 ABLATION STUDIES

In this section, we discuss the effectiveness of key contributions of our MAWM, that is, the adaptive motion-aware scheduler, the motion predictor, and the substitution of action-conditional video

prediction for image reconstruction. We randomly select 6 tasks for Atari 100k and 4 tasks for DeepMind Control Suite, the results of which are illustrated in Figure 4 and Figure 5, separately. For ablation studies on CBAM, harmonizers, and the choice of the autoencoder, please refer to Appendix F for more details.

**No AMAS** The green curve shows the performance of MAWM without the AMAS. On both benchmarks, we observe that the green curve always follows the blue curve or the red curve in each task, which demonstrates its adaptive control capability of scheduling the two predictors across diverse domains.

**No motion predictor** The motion predictor plays an essential role in environments where moving small objects matter, such as Breakout and Krull in Figure 4. While MAWM achieves a mean of 2.501, the HNS mean of six Atari tasks decreases to 2.110 without the motion predictor, which demonstrates the ability of the motion predictor to learn compact motion-aware representations in visual environments.

**With image reconstruction** Though MAWM is designed to learn representations via video prediction, we configure it to reconstruct images from posterior stochastic states and deterministic states. Under this configuration, we notice a sharp performance drop on the DeepMind Control Suite. Specifically, the average score of four tasks declines from 683.1 to 512.1, as shown in Figure 5. Our results suggest that visual representation learning via video prediction instead of image reconstruction is an important improvement for efficient policy learning. Obviously, it is only when RL agents understand the correlation of actions and resulting observations that they can predict satisfactory future frames.

## 5 CONCLUSION

In this paper, we have introduced MAWM, which is a general world model framework for visual MBRL that enables compact visual representation learning with a novel motion-aware mechanism. MAWM masters tasks across different domains for visual control, be it discrete or continuous. Specifically, compared with DreamerV3, MAWM achieves a 12% and 24% performance boost on average on Atari 100k benchmark and challenging tasks from DeepMind Control Suite, separately. Moreover, MAWM has established a new state-of-the-art result on these tasks for visual continuous control, even surpassing specialized model-free RL algorithms.

We identify three potential limitations of our work for future research. MAWM has difficulties in long-horizon video prediction, which is also the key problem in current MBRL methods (Alonso et al., 2024). Specifically, if the imagination step is large, predicted images may be incorrect in certain cases, even though predicted motion by MAWM remains accurate. Future work can try to find whether perfect long-horizon video prediction improves policy learning. Besides, although MAWM has been trained with fixed hyperparameters across domains, we currently train a standalone model for each task. An exciting avenue is to explore the potential of MAWM to finish different tasks within a model by effectively sharing common knowledge. Since MAWM learns task-specific relationships between actions and images, another promising avenue might be to integrate text-guided video generative models (Rombach et al., 2022; Wang et al., 2022; Brooks et al., 2023; Zhang et al., 2023a; Blattmann et al., 2023; Jeong et al., 2024; Luo et al., 2024) with world models. As text can be used to describe and aligned with actions, we believe this avenue can provide world models with more general ability.

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

## A DERIVATIONS

### A.1 INTERPRETATIONS OF REPRESENTATION MODEL AND DYNAMICS MODEL

Since the recurrent state space model for video prediction is a Markov process as shown in Figure 6, the encoder can be formulated as $q(s_{0:T-1}|a_{0:T-1}, o_{1:T}, o_0) = \prod_{t=0}^{T-1} q(s_t|o_{\leq t}, a_{<t}) = \prod_{t=0}^{T-1} q(s_t|o_t, h_t)$, where $h_t = h(s_{t-1}, a_{t-1})$ and $s_t = s(h_t, z_t)$ are deterministic functions. Therefore, the distribution of $s_t$ can be obtained if we know the distribution of the stochastic state $z_t$. We parameterize the distribution of $z_t$ via representation model $q_\phi(z_t|o_t, h_t)$, where $h_t = h(s_{t-1}, a_{t-1})$ can be implemented as a recurrent neural network. Similarly, we can obtain $p(s_t|s_{t-1}, a_{t-1})$ from $p(z_t|s_{t-1}, a_{t-1}) = p(z_t|h_t)$, which necessitates the dynamics model $p_\phi(\hat{z}_t|h_t)$. Furthermore, we have $D_{\text{KL}}(q(s_t|o_t, h_t)||p(s_t|h_t)) = D_{\text{KL}}(q(z_t|o_t, h_t)||p(z_t|h_t))$ due to our implementation of $s_t$, which is the concatenation of $h_t$ and $z_t$.

### A.2 PROOF OF EQUATION 7

Since we want to predict the next frame conditioned on the current state and action, the latent dynamics model is $p(o_{1:T}, s_{0:T-1}|a_{0:T-1}, o_0) = \prod_{t=1}^{T} p(o_t|s_{t-1}, a_{t-1}, o_0)p(s_{t-1}|s_{t-2}, a_{t-2}, o_0) = \prod_{t=1}^{T} p(o_t|s_{t-1}, a_{t-1})p(s_{t-1}|s_{t-2}, a_{t-2})$, where $p(s_0|s_{-1}, a_{-1})$ is defined as $p(s_0|o_0)$. Accordingly, the variational posterior is $q(s_{0:T-1}|a_{0:T-1}, o_{1:T}, o_0) = \prod_{t=0}^{T-1} q(s_t|o_{\leq t}, a_{<t})$, where we define $q(s_0|o_{\leq 0}, a_{<0})$ as $q(s_0|o_0)$. Using importance weighting and Jensen's inequality, the ELBO of the likelihood of the image conditioned on the first frame and history of actions is:

$$\ln p(o_{1:T}|a_{0:T-1}, o_0) \triangleq \ln \mathbb{E}_{p(s_{0:T-1}|a_{0:T-1}, o_0)} \left[ \prod_{t=1}^{T} p(o_t|s_{t-1}, a_{t-1}) \right]$$

$$= \ln \mathbb{E}_{q(s_{0:T-1}|a_{0:T-1}, o_0)} \left[ \frac{\prod_{t=1}^{T} p(o_t|s_{t-1}, a_{t-1})p(s_{t-1}|s_{t-2}, a_{t-2})}{q(s_{0:T-1}|a_{0:T-1}, o_0)} \right]$$

$$= \ln \mathbb{E}_{q(s_{0:T-1}|a_{0:T-1}, o_0)} \left[ \prod_{t=1}^{T} p(o_t|s_{t-1}, a_{t-1})p(s_{t-1}|s_{t-2}, a_{t-2})/q(s_{t-1}|o_{<t}, a_{<t-1}) \right]$$

$$\geq \mathbb{E}_{q(s_{0:T-1}|a_{0:T-1}, o_0)} \left[ \sum_{t=1}^{T} \ln p(o_t|s_{t-1}, a_{t-1}) + \ln p(s_{t-1}|s_{t-2}, a_{t-2}) - \ln q(s_{t-1}|o_{<t}, a_{<t-1}) \right]$$

$$= \sum_{t=1}^{T} \mathbb{E}_{q(s_{t-1}|o_{<t}, a_{<t-1})} [\ln p(o_t|s_{t-1}, a_{t-1})]$$

$$- \sum_{t=1}^{T} \mathbb{E}_{q(s_{t-2}|o_{<t-1}, a_{<t-2})} [D_{\text{KL}}(q(s_{t-1}|o_{<t}, a_{<t-1})||p(s_{t-1}|s_{t-2}, a_{t-2}))].$$

For $T \to \infty$, we always minimize the KL divergence of the latent dynamics models $p(s_{t-1}|s_{t-2}, a_{t-2})$ and the variational posterior $q(s_{t-1}|o_{<t}, a_{<t-1})$. Set $t' = t - 1$ and then substitute $t'$ for $t$. The second term will be

$$\sum_{t=0}^{\infty} \mathbb{E}_{q(s_{t-1}|o_{<t}, a_{<t-1})} [D_{\text{KL}}(q(s_t|o_{\leq t}, a_{<t})||p(s_t|s_{t-1}, a_{t-1}))].$$

We sample a batch from episodes and it would be helpful to minimize the KL divergence if we wish to have a better prediction of the next frame from other batches. Therefore, the modified objective is to maximize

$$\sum_{t=1}^{T} \left( \mathbb{E}_{q(s_{t-1}|o_{<t}, a_{<t-1})} [\ln p(o_t|s_{t-1}, a_{t-1}) - D_{\text{KL}}(q(s_t|o_{\leq t}, a_{<t})||p(s_t|s_{t-1}, a_{t-1}))] \right)$$

$$- D_{\text{KL}}(q(s_0|o_0)||p(s_0|o_0)).$$

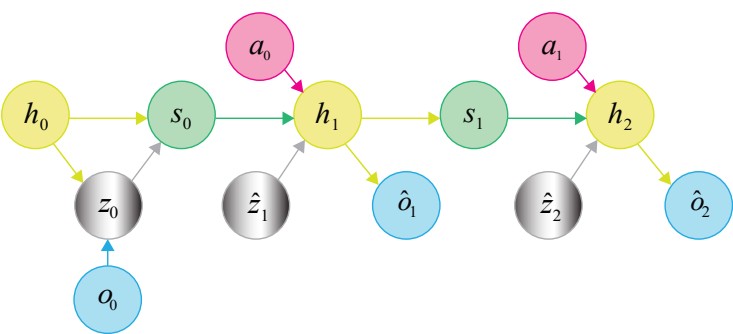

Figure 6: RSSM with video prediction.

# B MAWM ARCHITECTURE

## B.1 REPRESENTATION MODEL

The representation model consists of an image encoder and a representation predictor. Table 3 shows the process of the image encoder to obtain embeddings of an image $e_t$. Following that, the representation predictor takes as input $e_t$ and imagined deterministic hidden state $h_t$ to obtain the posterior stochastic hidden state $z_t$, as described in Table 4. LayerNorm denotes layer normalization (Ba, 2016), and SiLU is short for sigmoid-weighted linear units, an activate function which is formulated as $\text{SiLU}(x) = \frac{x}{1+e^{-x}}$. For clarity, we provide detailed descriptions of the implementation of the CBAM (Convolutional Block Attention Modul) stage.

Table 3: Structure of the image encoder

| Stage name | Output size | Submodule |
|---|---|---|
| CBAM | $64 \times 64$ | ChannelAttention, $r = 1$
SpatialAttention, $k = 3$ |
| Conv1 | $32 \times 32$ | $4 \times 4$, 32, stride 2
LayerNorm + SiLU |
| Conv2 | $16 \times 16$ | $4 \times 4$, 64, stride 2
LayerNorm + SiLU |
| Conv3 | $8 \times 8$ | $4 \times 4$, 128, stride 2
LayerNorm + SiLU |
| Conv4 | $4 \times 4$ | $4 \times 4$, 256, stride 2
LayerNorm + SiLU |
| CBAM | $4 \times 4$ | ChannelAttention, $r = 2$
SpatialAttention, $k = 1$ |
| Flatten | 4096 | the embedding of image $e_t$ |

For feature map $F_l \in \mathbb{R}^{C \times H \times W}$ in the $l$th layer of the encoder network, the output of the ChannelAttention submodule $F_l^c$ is

$$F_l^c = \sigma(W_2 \times ReLU(W_1 \times AvgPool(F_l)) + W_2 \times ReLU(W_1 \times MaxPool(F_l^c))) \odot F_l, \quad (9)$$

where $AvgPool(\cdot)$ and $MaxPool(\cdot)$ stand for adaptive average-pooling and max-pooling operations, respectively. $W_1 \in \mathbb{R}^{C/r \times C}$ and $W_2 \in \mathbb{R}^{C \times C/r}$ are weights of fully-connected layers, where $r$ is the reduction ratio. $ReLU$ represents the ReLU activation function (Glorot et al., 2011) after $W_1$. $\sigma$ denotes the sigmoid function and $\odot$ denotes element-wise multiplication. Similarly, the spatial attention submodule takes $F_l^c$ as inputs and the outputs are

$$F_s^c = \sigma(Conv_k([ChAvg(F_l^c), ChMax(F_l^c)])) \odot F_l^c, \quad (10)$$

Table 4: Structure of the representation predictor

| Stage name | Output size | Submodule |
|---|---|---|
| Inputs | $4096 + N_\text{deter}$ | Concatenate $h_t$ and $e_t$ |
| FC1 | $N_\text{hid}$ | Linear
LayerNorm + SiLU |
| FC2 | $Z_\text{num} \times Z_\text{class}$ | Linear
LayerNorm + SiLU
Reshape |

Table 5: Structure of the video predictor

| Stage name | Output size | Submodule |
|---|---|---|
| Inputs | $N_\text{stoch} + N_\text{deter}$ | Concatenate $h_t$ and $\hat{z}_t$ |
| FC1 | $4 \times 4$ | Linear
Reshape into tensors of 256 channels |
| Deconv1 | $8 \times 8$ | $4 \times 4$, 128, stride 2
LayerNorm + SiLU |
| Deconv2 | $16 \times 16$ | $4 \times 4$, 64, stride 2
LayerNorm + SiLU |
| Deconv3 | $32 \times 32$ | $4 \times 4$, 32, stride 2
LayerNorm + SiLU |
| Deconv4 | $64 \times 64$ | $4 \times 4$, 3, stride 2 |

where $ChAvg(\cdot)$ and $ChMax(\cdot)$ calculate the mean and the maximum value across channels of the feature map. The results of the above two operations are concatenated and convolved with the filters of size $k \times k$.

### B.2 PREDICTORS

**Predictors in decoder structure** Both the video predictor and motion predictor are expected to output tensors of height and width $64 \times 64$. To that end, we implement similar decoder networks for video and motion prediction, as depicted in Table 5 and Table 6.

Table 6: Structure of the motion predictor

| Stage name | Output size | Submodule |
|---|---|---|
| Inputs | $2N_\text{stoch} + N_\text{deter}$ | Concatenate $h_t$ and $\hat{s}_t$ |
| FC1 | $4 \times 4$ | Linear
Reshape into tensors of 256 channels |
| Deconv1 | $8 \times 8$ | $4 \times 4$, 128, stride 2
LayerNorm + SiLU |
| Deconv2 | $16 \times 16$ | $4 \times 4$, 64, stride 2
LayerNorm + SiLU |
| Deconv3 | $32 \times 32$ | $4 \times 4$, 32, stride 2
LayerNorm + SiLU |
| Deconv4 | $64 \times 64$ | $4 \times 4$, 1, stride 2
Sigmoid |

**Predictors for scalars** To enable agents to learn to behave well, it is necessary to predict reward and continuation flags. Table 7 displays MLP structures of reward predictor and continue predictor.

Table 7: Reward predictor and continue predictor

| Details of MLP | Reward predictor | continue predictor |
|---|---|---|
| Inputs | $N_{\text{stoch}} + N_{\text{deter}}$ | $N_{\text{stoch}} + N_{\text{deter}}$ |
| Hidden units | $N_{\text{unit}}$ | $N_{\text{unit}}$ |
| Outputs units | 255 | 1 |
| Activation function | SiLU | SiLU |
| Normalization | LayerNorm | LayerNorm |
| Layers | 2 | 2 |

## C ALGORITHM

The training process of MAWM is sketched out in Algorithm 1.

---
**Algorithm 1** MAWM Training

---
**Input:** An initialized replay buffer $\mathcal{D}$
**repeat**
  $o_0, r_0, c_0 \leftarrow$ `env.reset()`
  Initialize parameters of GMM using $o_0$ (Section 3.2)
  **for** $t = 0$ **to** MAX_STEP **do**
    $a_t \sim \pi(a_t | o_{\leq t}, a_{<t})$
    $o_{t+1}, r_{t+1}, c_{t+1} \leftarrow$ `env.step()`
    $m_{t+1} \leftarrow$ `GMM.predict(`$o_{t+1}$`)`
    **if** $c_{t+1} = 0$ **then**
      $t_m = t + 1$
      **break**
    **end if**
    Sample $B$ data of length $T$ from $\mathcal{D}$
    Encode images:$\{e_t\}_{t=k}^{k+T-1} \leftarrow$ `Image Encoder(`$\{o_t\}_{t=k}^{k+T-1}$`)`
    Predict $\{\hat{o}_t, \hat{m}_t, \hat{z}_t, z_t, \hat{r}_t, \hat{c}_t\}_{t=k}^{k+T-1}$(Formula 1)
    Compute total loss $\mathcal{L}(\phi, \sigma)$(Formula 2, 3, 6, 8)
    Update parameters $\phi$ and $\sigma$
    Actor-critic learning in imagined trajectories
  **end for**
  $\mathcal{D}$`.add(`$\{o_t, m_t, a_{t-1}, r_t, c_t\}_{t=0}^{t_m}$`)`
**until** Training is stopped

---

## D HYPERPARAMETERS

### D.1 MAWM

Table 8 shows hyperparameters of MAWM. These hyperparameters are fixed on both Atari 100k and DeepMind Control benchmarks.

### D.2 DREAMERV3

We used the default parameters and reproduced the results based on the implementation of DreamerV3 in PyTorch, which performs slightly better than the original implementation of DreamerV3 in TensorFlow (Hafner et al., 2023).

Table 8: Hyperparameters in our world model, MAWM. $N_{\text{stoch}}$ de facto denotes the dimension of the flattened version of $z_t$. That is to say, $N_{\text{stoch}} = Z_{\text{num}} \cdot Z_{\text{class}}$ for discrete representations, which is our choice. $N_{\text{stoch}} = Z_{\text{num}}$ when continuous representations are applied.

| Type | Hyperparameter | Value |
|---|---|---|
| General | Image size | $64 \times 64 \times 3$ |
| | Batch size | 16 |
| | Batch length $T$ | 64 |
| | Gradient Clipping | 1000 |
| | Discount factor $\gamma$ | 0.997 |
| | Lambda $\lambda$ | 0.95 |
| World model | Number of stochastic variables $Z_{\text{num}}$ | 64 |
| | Classes per stochastic variable $Z_{\text{class}}$ | 32 |
| | Number of deterministic units $N_{\text{deter}}$ | 512 |
| | Number of stochastic units $N_{\text{stoch}}$ | 2048 |
| | Number of MLP units $N_{\text{unit}}$ | 512 |
| | Number of RSSM units $N_{\text{hid}}$ | 512 |
| | Imagination horizon | 15 |
| | First frame prediction | False |
| | Motion prediction | True |
| | Ratio of motion-aware region $r_{\text{dizzy}}$ | 0.05 |
| | Updatation per interaction | 1 |
| | Harmonizers | True |
| | Optimizer | AdamW (Loshchilov, 2017) |
| | AdamW episilon $\epsilon$ | $1 \times 10^{-8}$ |
| | AdamW betas $(\beta_1, \beta_2)$ | $(0.9, 0.999)$ |
| | Learning rate | $1 \times 10^{-4}$ |
| | Gradient clipping | 1000 |
| Loss term | Video prediction weight $\beta_{\text{o}}$ | 1.0 |
| | Motion prediction weight $\beta_{\text{m}}$ | 0.5 |
| | Reward prediction weight $\beta_{\text{r}}$ | 1.0 |
| | Continuation flags prediction weight $\beta_{\text{c}}$ | 1.0 |
| | Dynamics weight $\beta_{\text{dyn}}$ | 0.5 |
| | Representation weight $\beta_{\text{rep}}$ | 0.1 |
| | Focal loss alpha $\alpha$ | 0.15 |
| | Focal loss gamma $\gamma$ | 4 |
| | Motion-aware weight $\omega$ | 0.5 |

### D.3 HARMONYDREAM

Since no hyperparameter is introduced in HarmonyDream (Ma et al., 2024a), we implemented harmonizers following recommendations from the authors. We reproduced the results based on the aforementioned implementation of DreamerV3 with harmonious loss, as suggested by the authors in their articles.

## D.4 TD-MPC2

Results for seven out of eight tasks from DeepMind Control Suite can be found at the official repository in Github, except Hopper Hop. We follow the official implementation of TD-MPC2 (Hansen et al., 2024), use the default hyperparameters, and select the default 5M parameters for the single task.

## E   ADDITIONAL RESULTS ON ATARI 100K

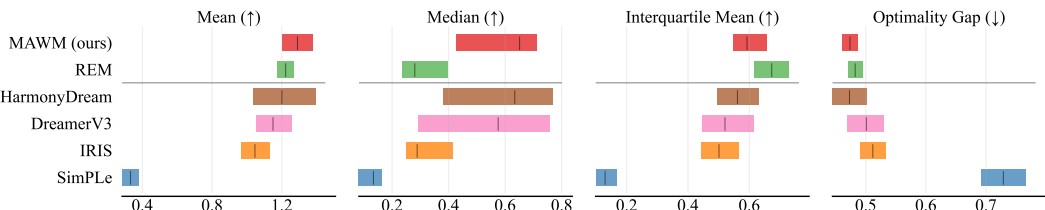

Figure 7: Mean, median, and inter-quantile mean (IQM) human-normalized scores and the optimality gap (Agarwal et al., 2021) with 95% stratified bootstrap confidence intervals on the Atari 100k benchmark.

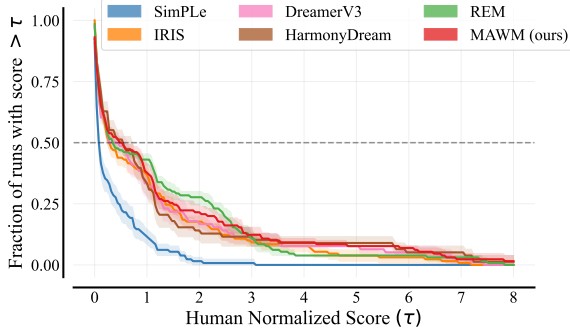

Figure 8: Performance profiles (Agarwal et al., 2021). The curve of each algorithm shows the proportion of runs in which human-normalized scores are greater than the given score threshold.

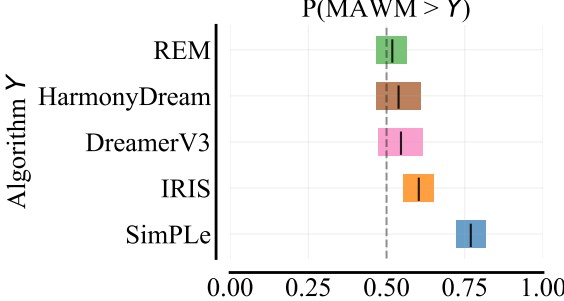

Figure 9: Each row represents the probability of improvement (Agarwal et al., 2021) that our algorithm outperforms the corresponding baseline in a randomly selected task from all tasks with 95% stratified bootstrap confidence intervals.

# F ADDITIONAL ABLATION STUDIES

## F.1 CBAM AND HARMONIZERS

We conduct additional ablation studies on CBAM and Harmonizers to study the function of both modules on the Atari 100k benchmark. Table 9 demonstrates that MAWN without both modules still attains a mean human-normalized score of 1.289, outperforming the best baseline, REM, which has a mean human-normalized score of 1.222. Although the average performance of MAWM without both modules is very close to MAWM with the standard configuration, MAWM with the standard configuration performs more consistently on all tasks.

Table 9: Ablation studies on CBAM and Harmonizers on the Atari 100k benchmark. Both: CBAM and Harmonizers, Standard: standard configurations of MAWM in the body of our paper.

| Game | REM | MAWM(Ours) | | |
| --- | --- | --- | --- | --- |
| | | - Both | - CBAM | Standard |
| Alien | 607.2 | 1089.0 | **1165.4** | 776.4 |
| Amidar | 95.3 | **210.9** | 110.8 | 144.2 |
| Assault | **1764.2** | 1075.1 | 790.9 | 883.4 |
| Asterix | **1637.5** | 1466.3 | 1201.8 | 1096.9 |
| BankHeist | 19.2 | 517.2 | **987.5** | 742.6 |
| BattleZone | 11826 | 8060.0 | 10696.7 | **13372.0** |
| Boxing | **87.5** | 80.9 | 84.2 | 85.4 |
| Breakout | 90.7 | **108.7** | 40.6 | 71.8 |
| ChopperCommand | **2561.2** | 899.0 | 818.0 | 904.0 |
| CrazyClimber | 76547.6 | 82506.7 | **89538.3** | 89038.6 |
| DemonAttack | **5738.6** | 149.1 | 157.4 | 152.2 |
| Freeway | **32.3** | 0.0 | 0.0 | 0.0 |
| Frostbite | 240.5 | 2040.0 | **2449.2** | 692.6 |
| Gopher | 5452.4 | 3403.1 | **8012.3** | 4415.8 |
| Hero | 6484.8 | **11482.4** | 8139.8 | 8801.8 |
| Jamesbond | 391.2 | **477.0** | 376.3 | 337.2 |
| Kangaroo | 467.6 | 1726.7 | 1836.0 | **3875.6** |
| Krull | 4017.7 | 8312.8 | 8408.5 | **8729.6** |
| KungFuMaster | **25172.2** | 19122.7 | 21415.3 | 23434.6 |
| MsPacman | 962.5 | 1557.3 | 1573.7 | **1580.7** |
| Pong | 18.0 | **20.2** | 18.3 | 20.1 |
| PrivateEye | 99.6 | **3288.6** | 1423.8 | -472.5 |
| Qbert | 743 | **4237.2** | 1145.1 | 1664.4 |
| RoadRunner | 14060.2 | **20635.7** | 14725.3 | 12518.6 |
| Seaquest | **1036.7** | 440.0 | 554.0 | 557.9 |
| UpNDown | 3757.6 | 15716.1 | 15952.4 | **28408.2** |
| #Superhuman($\uparrow$) | **12** | 10 | 11 | **12** |
| Mean($\uparrow$) | 1.222 | 1.289 | 1.258 | **1.290** |
| Median($\uparrow$) | 0.280 | 0.512 | 0.578 | **0.651** |

## F.2 CHOICE OF THE AUTOENCODER

To further explore whether our variational autoencoder for video is a better choice than a masked autoencoder (MAE; He et al., 2022) for image reconstruction, we also utilize a masking strategy

on the image embeddings encoded by the representation model in the same way as MWM (Seo et al., 2023). Specifically, we disentangle training of the representation model from training of MAWM and input frozen image embeddings without masking to the dynamics model. We denote the resulting world model without the AMAS and motion predictor as MAE with a masking ratio of 75%, as suggested by Seo et al. (2023). Results in Table 10 demonstrate that our variational autoencoder for video ensures consistent excellent performance on tasks from DeepMind Control. Furthermore, the AMAS and the motion predictor are instrumental in enhancing compact visual representation learning for MAE.

Table 10: Ablation studies on VAE for video on eight challenging tasks from DeepMind Control Suite. AMASMO: AMAS and motion predictor.

| Task | TD-MPC2 | MAE | MAE + AMASMO | MAWM(ours) |
|---|---|---|---|---|
| Acrobot Swingup | 295.3 | 236.6 | 416.1 | **452.1** |
| Cartpole Swingup Sparse | **790.0** | 472.9 | 548.7 | 666.7 |
| Cheetah Run | 537.3 | 565.7 | 765.3 | **874.3** |
| Finger Turn Hard | 885.2 | 433.4 | 856.5 | **935.0** |
| Hopper Hop | 302.9 | 52.5 | **399.3** | 311.5 |
| Quadruped Run | 283.1 | **860.3** | 537.0 | 648.7 |
| Quadruped Walk | 323.5 | **883.7** | 835.3 | 580.3 |
| Reacher hard | **909.6** | 705.0 | 627.3 | 654.9 |
| Mean(↑) | 540.9 | 526.3 | 623.2 | **640.4** |
| Median(↑) | 430.4 | 519.3 | 588.0 | **651.8** |

# G ATARI 100K CURVES

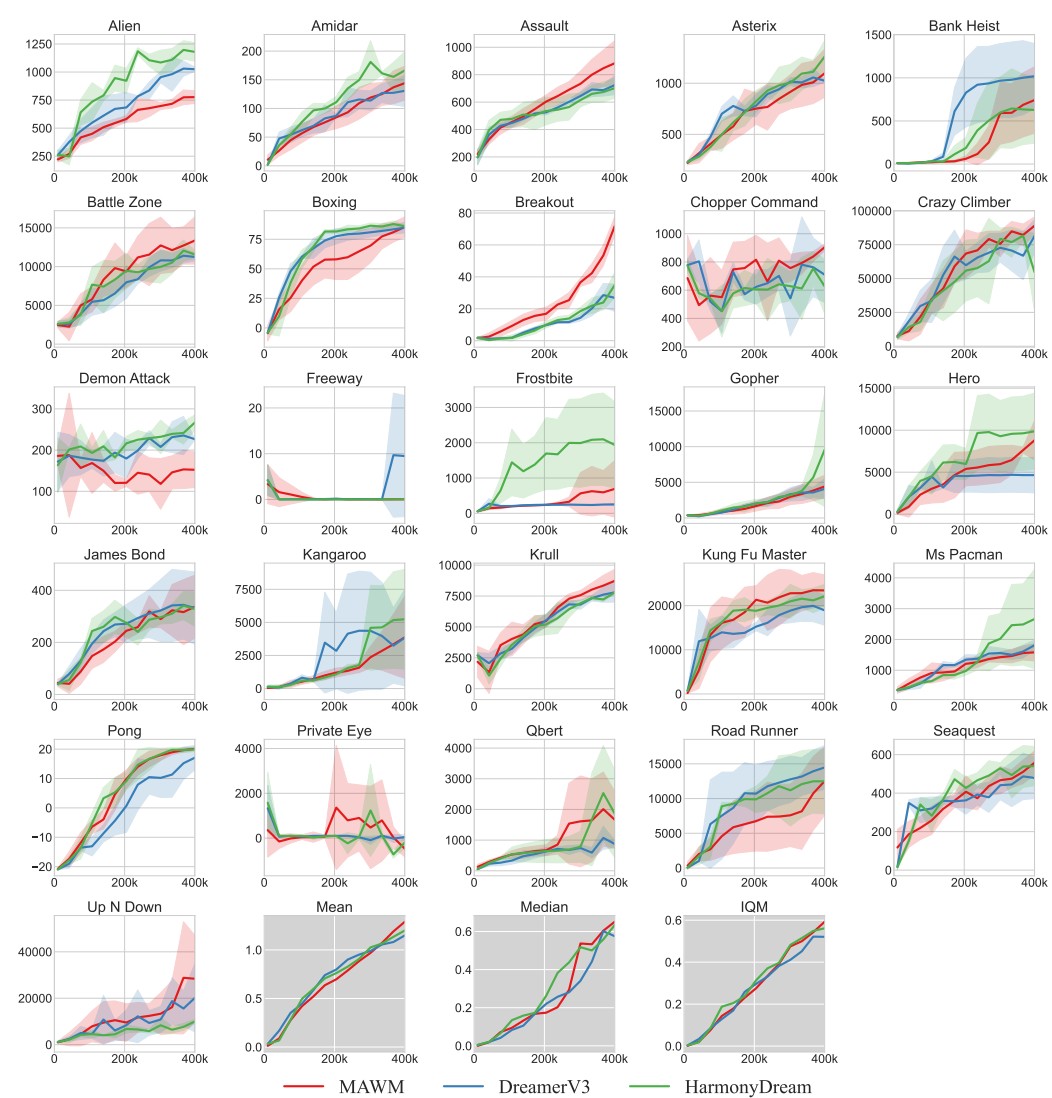

Figure 10: Training curves of MAWM and DreamerV3 on the Atari 100k benchmark. 100k interaction data amounts to 400k frames.

## H DeepMind Control Curves

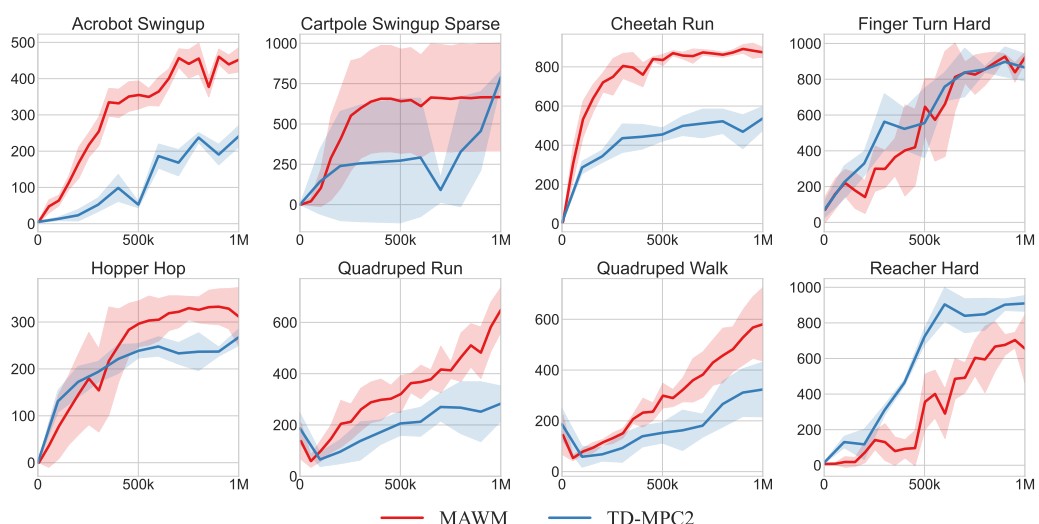

Figure 11: Training curves of MAWM and TD-MPC2 on the challenging tasks from DeepMind Control. 1M frames corresponds to 500k interaction data.

## I Computational Resources

MAWM consists of 45M parameters. We report our results for each task on Atari 100k and Deep-Mind Control Suite based on experiments over 5 random seeds. Experiments on Atari 100k were conducted with NVIDIA V100 32GB GPUs. Training on Atari 100k, with three tasks running on the same GPU in parallel, took about 1.2 days, resulting in an average of 0.4 days per environment. Experiments on DMC were conducted with NVIDIA GeForce RTX 4090 24GB GPUs. Training on DMC, with three tasks running on the same GPU in parallel, took 1.8 days, resulting in an average of 0.6 days per environment. As a reference, DIAMOND (Alonso et al., 2024) took approximately 2.9 days on a single NVIDIA GeForce RTX 4090 for training on a task of Atari 100k.

## J Broader Comparisons on Atari 100k

Table 11 showcases MBRL methods with lookahead search, including EfficientZero V2 (Wang et al., 2024), the state-of-the-art MBRL method on the Atari 100k benchmark. We here exclude DIAMOND (Alonso et al., 2024) because it relies on the video generation quality of the diffusion model, which is out of the scope of this study.

## K Extended results on DeepMind Control Suite

For a more comprehensive evaluation of MAWM, we conducted an extensive experiment on all the 20 tasks from DeepMind Control Suite. As demonstrated in Table 12, MAWM has set a state-of-the-art result on the DeepMind Control Suite. Moreover, MAWM achieves the highest scores on half of the tasks among the baselines and performs consistently well.

Table 11: Game scores and human normalized aggregate metrics on the Atari 100k benchmark with MBRL methods. We highlight the highest and the second highest scores among all baselines in bold and with underscores, respectively.

| Game | Random | Human | Lookahead search | | No lookahead search | | | | | | |
|---|---|---|---|---|---|---|---|---|---|---|---|
| | | | MuZero | EZ-V2 | SimPLe | IRIS | DreamerV3 | STORM | HarmoyDream | REM | Ours |
| Alien | 227.8 | 7127.7 | 530.0 | **1557.7** | 616.9 | 420.0 | 1024.9 | 983.6 | _1179.3_ | 607.2 | 776.4 |
| Amidar | 5.8 | 1719.5 | 38.8 | _184.9_ | 88.0 | 143.0 | 130.8 | **204.8** | 166.3 | 95.3 | 144.2 |
| Assault | 222.4 | 742.0 | 500.1 | _1757.5_ | 527.2 | 1524.4 | 723.6 | 801.0 | 701.7 | **1764.2** | 883.4 |
| Asterix | 210.0 | 8503.3 | 1734.0 | **61810.0** | 1128.3 | 853.6 | 1024.2 | 1260.2 | 1260.2 | _1637.5_ | 1096.9 |
| BankHeist | 14.2 | 753.1 | 192.5 | **1316.7** | 753.1 | 53.1 | _1018.9_ | 641.2 | 627.1 | 19.2 | 742.6 |
| BattleZone | 2360.0 | 37187.5 | 7687.5 | **14433.3** | 5184.4 | 13074.0 | 11246.7 | _13540.0_ | 11563.3 | 11826 | 13372.0 |
| Boxing | 0.1 | 12.1 | 15.1 | 75.0 | 9.1 | 70.1 | 84.8 | 79.7 | _86.0_ | **87.5** | 85.4 |
| Breakout | 1.7 | 30.5 | 48.0 | **400.1** | 16.4 | 83.7 | 26.9 | 15.9 | 34.9 | _90.7_ | 71.8 |
| ChopperCommand | 811.0 | 7387.8 | 1350.0 | 1196.6 | 1246.9 | 1565.0 | 709.7 | _1888.0_ | 627.0 | **2561.2** | 904.0 |
| CrazyClimber | 10780.5 | 35829.4 | 56937.0 | 112363.3 | 35829.4 | 59324.2 | 81414.7 | 66776.0 | 54687.3 | _76547.6_ | **89038.6** |
| DemonAttack | 152.1 | 1971.0 | 3527.0 | **22773.5** | 1971.0 | 2034.4 | 226.5 | 164.6 | 267.0 | _5738.6_ | 152.2 |
| Freeway | 0.0 | 29.6 | 21.8 | 0.0 | 20.3 | _31.1_ | 9.5 | 0.0 | 0.0 | **32.3** | 0.0 |
| Frostbite | 65.2 | 4334.7 | 255.0 | _1136.3_ | 254.7 | 259.1 | 251.7 | 1316.0 | **1937.9** | 240.5 | 692.6 |
| Gopher | 257.6 | 2412.5 | 1256.0 | 3868.7 | 771.0 | 2236.1 | 4074.9 | _8239.6_ | **9564.7** | 5452.4 | 4415.8 |
| Hero | 1027.0 | 30826.4 | 3095.0 | 9705.0 | 2656.6 | 7037.4 | 4650.9 | **11044.3** | _9865.3_ | 6484.8 | 8801.8 |
| Jamesbond | 29.0 | 302.8 | 87.5 | _468.3_ | 125.3 | 462.7 | 331.8 | **509.0** | 327.8 | 391.2 | 337.2 |
| Kangaroo | 52.0 | 3035.0 | 62.5 | 1886.7 | 323.1 | 838.2 | 3851.7 | _4208.0_ | **5237.3** | 467.6 | 3875.6 |
| Krull | 1598.0 | 2665.5 | 4890.8 | **9080.0** | 4539.9 | 6616.4 | 7796.6 | 8412.6 | 7784.0 | 4017.7 | _8729.6_ |
| KungFuMaster | 258.5 | 22736.3 | 18813.0 | **28883.3** | 258.5 | 21759.8 | 18917.1 | 26182.0 | 22131.7 | 25172.2 | 23434.6 |
| MsPacman | 307.3 | 6951.6 | 1265.6 | 2251.0 | **6951.6** | 999.1 | 1813.3 | _2673.5_ | 2663.3 | 962.5 | 1580.7 |
| Pong | -20.7 | 14.6 | -6.7 | **20.8** | 12.8 | 14.6 | 17.1 | 11.3 | 20.0 | 18 | _20.1_ |
| PrivateEye | 24.9 | 69571.3 | 56.3 | 99.8 | **69571.3** | 100.0 | 47.4 | _7781.0_ | -198.6 | 99.6 | -472.5 |
| Qbert | 163.9 | 13455.0 | 3952.0 | **16058.3** | 1288.8 | 745.7 | 873.2 | _4522.5_ | 1863.3 | 743 | 1664.4 |
| RoadRunner | 11.5 | 7845.0 | 2500.0 | **27516.7** | 7845.0 | 9614.6 | 14478.3 | _17564.0_ | 12478.3 | 14060.2 | 12518.6 |
| Seaquest | 68.4 | 42054.7 | 208.0 | **1974.0** | 683.3 | 661.3 | 479.1 | 525.2 | 540.7 | _1036.7_ | 557.9 |
| UpNDown | 533.4 | 11693.2 | 2896.9 | 15224.3 | 533.4 | 3546.2 | _20183.2_ | 7985.0 | 10007.1 | 3757.6 | **28408.2** |
| #Superhuman(↑) | 0 | N/A | 5 | **15** | 1 | 10 | 10 | 9 | 9 | _12_ | _12_ |
| Mean(↑) | 0.000 | 1.000 | 0.562 | **2.428** | 0.322 | 1.046 | 1.150 | 1.222 | 1.200 | 1.222 | _1.290_ |
| Median(↑) | 0.000 | 1.000 | 0.227 | **1.286** | 0.134 | 0.289 | 0.575 | 0.425 | 0.634 | 0.280 | _0.651_ |
| IQM(↑) | 0.000 | 1.000 | N/A | N/A | 0.130 | 0.501 | 0.521 | 0.561 | 0.561 | **0.673** | _0.593_ |
| Optimality gap(↓) | 1.000 | 0.000 | N/A | N/A | 0.729 | 0.512 | 0.501 | **0.472** | _0.473_ | 0.482 | 0.474 |

Table 12: Scores achieved for all the 20 tasks from DeepMind Control Suite with a budget of 500k interactions. We highlight the highest and the second highest scores among all baselines in bold and with underscores, respectively.

| Task | CURL | DrQ-v2 | DreamerV3 | TD-MPC2 | MAWM (Ours) |
|---|---|---|---|---|---|
| Acrobot Swingup | 5.1 | 128.4 | 210.0 | _241.3_ | **452.1** |
| Cartpole Balance | 979.0 | 991.5 | _996.4_ | 993.0 | **999.4** |
| Cartpole Balance Sparse | 981.0 | 996.2 | **1000.0** | **1000.0** | **1000.0** |
| Cartpole Swingup | 762.7 | _858.9_ | 819.1 | 831.0 | **871.4** |
| Cartpole Swingup Sparse | 236.2 | 706.9 | **792.9** | _790.0_ | 666.7 |
| Cheetah Run | 474.3 | 691.0 | _728.7_ | 537.3 | **874.3** |
| Cup Catch | 965.5 | 931.8 | _957.1_ | 917.5 | **966.9** |
| Finger Spin | 877.1 | 846.7 | _818.5_ | **984.9** | 596.7 |
| Finger Turn Easy | 338.0 | 448.4 | 787.7 | _820.8_ | **916.6** |
| Finger Turn Hard | 215.6 | 220.0 | 810.8 | _865.6_ | **935.0** |
| Hopper Hop | 152.5 | 189.9 | **369.6** | 267.6 | _311.5_ |
| Hopper Stand | 786.8 | 893.0 | _900.6_ | 790.3 | **926.2** |
| Pendulum Swingup | 376.4 | **839.7** | 806.3 | 832.6 | _835.0_ |
| Quadruped Run | 141.5 | _407.0_ | 352.3 | 283.1 | **648.7** |
| Quadruped Walk | 123.7 | **660.3** | 352.6 | 323.5 | _580.3_ |
| Reacher Easy | 609.3 | 910.2 | 898.9 | **982.2** | _937.7_ |
| Reacher Hard | 400.2 | 572.9 | 499.2 | **909.6** | _654.9_ |
| Walker Run | 376.2 | 517.1 | _757.8_ | 671.9 | **784.8** |
| Walker Stand | 463.5 | _974.1_ | **976.7** | 878.1 | 966.6 |
| Walker Walk | 828.8 | 762.9 | **955.8** | 939.6 | _942.6_ |
| Mean(↑) | 504.7 | 677.4 | 739.6 | _743.0_ | **793.4** |
| Median(↑) | 431.8 | 734.9 | 808.5 | _831.8_ | **872.8** |

## L   DMC-GB2

**DMC-GB2** (Almuzairee et al., 2024) is an extension of the DMControl Generalization Benchmark (Hansen & Wang, 2021), which consists of six continuous control tasks, *i.e.*, Cartpole Swingup, Cheetah Run, Cup Catch, Finger Spin, Walker Stand, and Walker Walk. It provides various test environments that are visually distinct from the training environment, as shown in Figure 12, and challenges RL agents to the ability of visual generalization. Specially designed algorithms, such as SVEA (Hansen et al., 2021) and SADA (Almuzairee et al., 2024) on the benchmark need pairs of original images and augmented images. In comparison, MAWM applies to the benchmark without any change. We train MAWM on DMC-GB2 with the same fixed hyperparameters over 5 random seeds. To evaluate the generalization ability of MAWM, we evaluate its performance on the whole Photometric Test Set. As shown in tables 13 to 18, MAWM is competitive with SADA, the state-of-the-art algorithm designed specifically for the benchmark. However, the comparison is unfair to our method since MAWM does not require original images and augmented images. Nevertheless, the generalization ability of MAWM on the DMC-GB2 benchmark indicates that MAWM has the potential to master a broader range of environments and work in a real-world application.

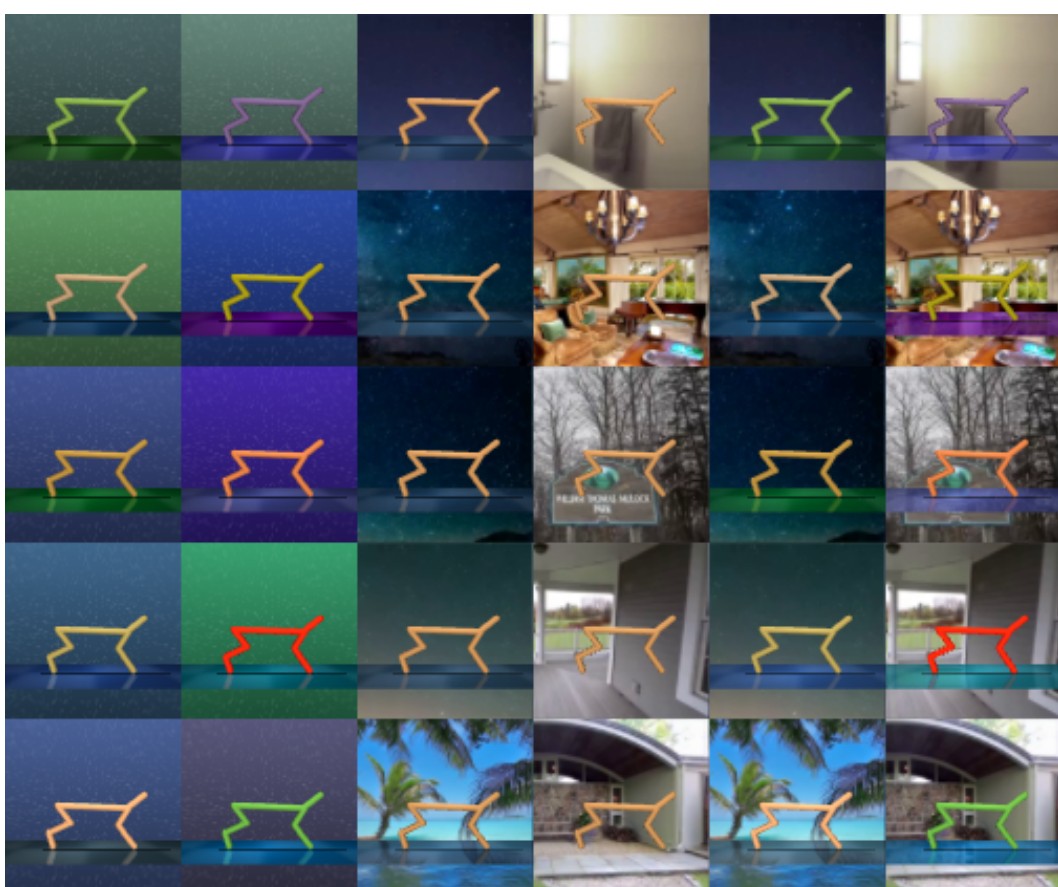

Figure 12: Snapshots of the Cheetah Run task in DMC-GB2 (Almuzairee et al., 2024). The test environments consist of Color Easy, Color Hard, Video, Video Hard, Color Video Easy, and Color Video Hard (from left to right). Color refers to environments with randomized colors while Video refers to the substitution of the original background for video from natural environments.

Table 13: Scores achieved in Color Easy test environments.

| Task | DrQ | SVEA | SADA | MAWM (Ours) |
|------|-----|------|------|-------------|
| Cartpole Swingup | 696 | 542 | 704 | **812** |
| Cheetah Run | 341 | 203 | 252 | **538** |
| Cup Catch | 833 | 821 | **969** | 954 |
| Finger Spin | 795 | **924** | 895 | 587 |
| Walker Stand | 826 | 900 | 965 | **970** |
| Walker Walk | 582 | 755 | **837** | 686 |
| Mean(↑) | 679 | 691 | **770** | 758 |

Table 14: Scores achieved in Color Hard test environments.

| Task | DrQ | SVEA | SADA | MAWM (Ours) |
|------|-----|------|------|-------------|
| Cartpole Swingup | 441 | 478 | 716 | **774** |
| Cheetah Run | 178 | 133 | 239 | **567** |
| Cup Catch | 520 | 779 | **961** | 914 |
| Finger Spin | 466 | 802 | **868** | 576 |
| Walker Stand | 527 | 861 | 963 | **964** |
| Walker Walk | 265 | 667 | **825** | 705 |
| Mean(↑) | 400 | 620 | **762** | 750 |

Table 15: Scores achieved in Video Easy test environments.

| Task | DrQ | SVEA | SADA | MAWM (Ours) |
|------|-----|------|------|-------------|
| Cartpole Swingup | 375 | 427 | 524 | **586** |
| Cheetah Run | 75 | 102 | 121 | **615** |
| Cup Catch | 523 | 736 | **934** | 691 |
| Finger Spin | 441 | 774 | **875** | 512 |
| Walker Stand | 603 | 945 | 923 | **969** |
| Walker Walk | 390 | 788 | **791** | 728 |
| Mean(↑) | 401 | 629 | **695** | 684 |

Table 16: Scores achieved in Video Hard test environments.

| Task | DrQ | SVEA | SADA | MAWM (Ours) |
|------|-----|------|------|-------------|
| Cartpole Swingup | 98 | 259 | 363 | **449** |
| Cheetah Run | 25 | 28 | 82 | **240** |
| Cup Catch | 111 | 416 | **662** | 288 |
| Finger Spin | 7 | 263 | **566** | 400 |
| Walker Stand | 154 | 429 | 702 | **872** |
| Walker Walk | 36 | 264 | 270 | **613** |
| Mean(↑) | 72 | 277 | 441 | **477** |

Table 17: Scores achieved in Color Video Easy test environments.

| Task | DrQ | SVEA | SADA | MAWM (Ours) |
|------|-----|------|------|-------------|
| Cartpole Swingup | 327 | 427 | 570 | **571** |
| Cheetah Run | 60 | 100 | 153 | **470** |
| Cup Catch | 447 | 716 | **931** | 645 |
| Finger Spin | 310 | 705 | **850** | 510 |
| Walker Stand | 487 | 852 | 945 | **960** |
| Walker Walk | 208 | 681 | **791** | 695 |
| Mean(↑) | 307 | 580 | **707** | 642 |

Table 18: Scores achieved in Color Video Hard environments.

| Task | DrQ | SVEA | SADA | MAWM (Ours) |
|------|-----|------|------|-------------|
| Cartpole Swingup | 94 | 294 | 426 | **437** |
| Cheetah Run | 26 | 44 | 99 | **531** |
| Cup Catch | 122 | 484 | **697** | 573 |
| Finger Spin | 2 | 307 | **633** | 398 |
| Walker Stand | 170 | 659 | 906 | **952** |
| Walker Walk | 42 | 421 | **686** | 648 |
| Mean(↑) | 76 | 368 | 575 | **590** |

# M    VIDEO PREDICTION ON ATARI 100K

Since current video generation models were not pre-trained with low-resolution images, we resize images to $512 \times 512$ as inputs for pre-trained video generation models. We tried several pre-trained video generation models (Blattmann et al., 2023; Esser et al., 2024) originating from Stable Diffusion (Rombach et al., 2022) to generate future frames conditioned on the past frames and proper prompts. As shown in Figure 13 and Figure 14, the pre-trained Stable Diffusion model often fails to catch the moving patterns of small targets, while MAWM can make fine-grained predictions of future frames.

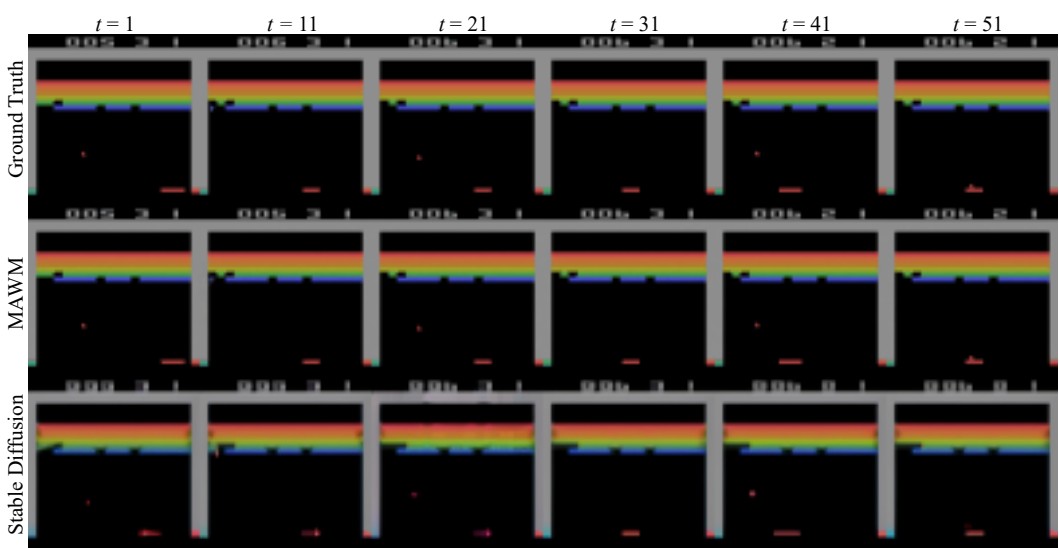

Figure 13: Comparison of predicted frames for the game Breakout by MAWM and Stable Diffusion (Rombach et al., 2022). Notably, at time $t = 11$, MAWM succeeds in predicting the change of score from 5 to 6 in the upper part of the frame and the color change of the tiny ball. However, the pre-trained Stable Diffusion model even misses information about the tiny ball at time $t = 51$.

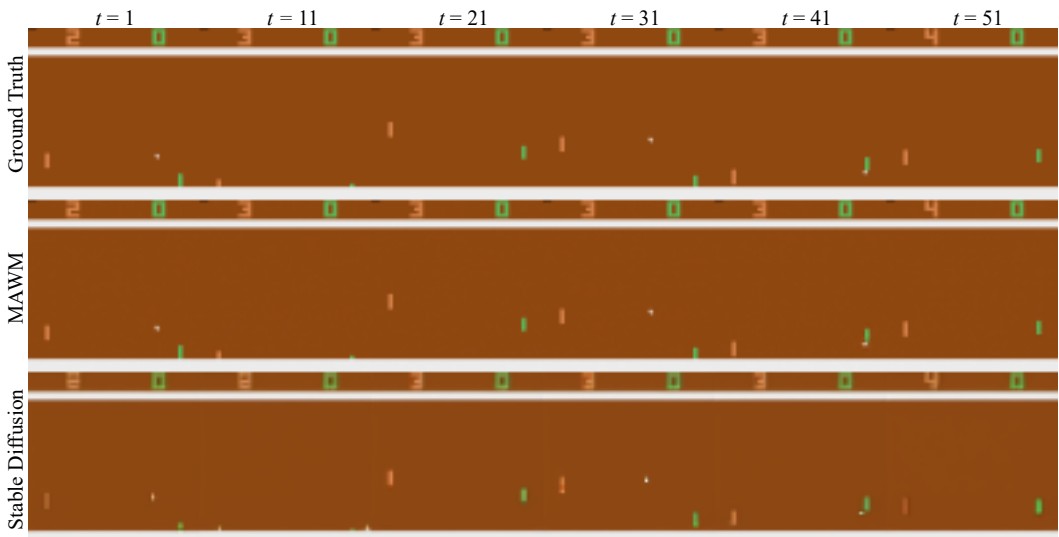

Figure 14: Comparison of predicted frames for the game Pong by MAWM and Stable Diffusion (Rombach et al., 2022). MAWM succeeds in predicting the change of score from 3 to 4 in the upper part of the frame at time $t = 11$ and has a more accurate estimation of the moving tiny objects than the pre-trained Stable Diffusion model.

## N  EXTENDED RELATED WORK

**Model-free visual reinforcement learning**  It has been a crucial challenge for reinforcement learning algorithms to learn policy from high-dimensional images. UNREAL (Jaderberg et al., 2017) showed the significance of auxiliary unsupervised objectives by achieving amazing scores on 57 games of Atari after 25M steps, averaging 880% mean human-normalized score. Following this work, several attempts (Gelada et al., 2019; Schwarzer et al., 2021; Yu et al., 2021) were made to train agents via predicting future latent states. After Oord et al. (2018) introduced Contrastive Predictive Coding (CPC), a general method that integrates video prediction with a probabilistic contrastive loss, called InfoNCE, contrastive representation learning method s (Anand et al., 2019; Mazoure et al., 2020; Laskin et al., 2020; Yarats et al., 2021a) were explored. To avoid issues of distraction from task-relevant elements, Deep Bisimulation for Control (DBC; Zhang et al., 2021) applied bisimulation metrics (Ferns & Precup, 2014; Castro, 2020) to learning representations that are invariant to task-irrelevant visual details. To enable robust learning directly from images instead of auxiliary loss, DrQ (Yarats et al., 2021b) proposed a data augmentation technique, which was incorporated and combined with linear decay for the variance of the exploration noise with DDPG (Lillicrap, 2015) algorithm in later DrQ-v2 (Yarats et al., 2022). By using the above techniques, DrQ-v2 established a strong baseline on the DMC benchmark for model-free RL algorithms.

**Moving object detection**  Real-world Applications such as video surveillance and optical motion capture, often require a moving object detection step to locate moving objects in a video. Therefore, moving object detection has attracted much attraction in recent decades (Kulchandani & Dangarwala, 2015). Approaches for moving object detection can be divided into three main categories: frame difference, optical flow, and background subtraction. Traditional frame difference methods (Jain & Nagel, 1979; Haritaoglu et al., 2000) employ pixel-wise difference between two successive frames. Optical flow methods (Horn & Schunck, 1981; Beauchemin & Barron, 1995) detect objects by establishing the optical flow field of images and calculating the motion vector of the associated pixels but their applications were limited by the significant computational demands (Agarwal et al., 2016; Shah & Xuezhi, 2021). Using semantic segmentation network (Ravi et al., 2024; Xie et al., 2024) to produce motion clues needs labeled data or extra demonstrations. Background subtraction is the most popular method (Chapel & Bouwmans, 2020) due to an excellent balance between robustness and computational overhead. The adaptive GMM method we employ in Section 3.2 falls in this category. We recommend comprehensive surveys (Bouwmans, 2014; Bouwmans et al., 2018; Chapel & Bouwmans, 2020; Kalsotra & Arora, 2022) for more details.

**Video Prediction**  Video prediction is to generate future frames based on existing video content. Current video prediction algorithms can be divided into three categories, *i.e.*, deterministic prediction, stochastic prediction, and generative prediction (Ming et al., 2024b). Algorithms that make deterministic prediction aims to perform pixel-level fitting based on deterministic models. PredNet (Lotter et al., 2016) pioneered the application of the recurrent convolutional network in video prediction. ConvLSTM (Shi et al., 2015) integrated LSTM with a convolutional neural network to proficiently capture spatiotemporal dynamics, which has a significant impact on subsequent video prediction models . (Xu et al., 2018; Wang et al., 2018; Gao et al., 2022; Straka et al., 2023). Several studies (Luc et al., 2017; Wu et al., 2020; Hu et al., 2023) incorporate additional information such as optical flow and semantic maps to enhance prediction quality. Qi et al. (2019) introduced a 3D motion decomposition module to predict ego-motion and foreground motion, which are then combined to generate a future 3D scene. With the predicted 3D scene, future frames are synthesized by projective transformations. However, deterministic algorithms often produce blurry images due to confining possible outcomes to fixed results (Oprea et al., 2020). To that end, several works introduced stochastic distributions into deterministic models (Kalchbrenner et al., 2017; Babaeizadeh et al., 2018) or leveraged probabilistic models (Mathieu et al., 2015; Lee et al., 2018). MOSO (Sun et al., 2023)is a notable approach that addresses the problem of dynamic background shifts via motion, scene, and object decomposition under a two-stage framework. It first utilizes the VQVAE Van Den Oord et al. (2017) to learn token-level representations via an image reconstruction task and then employs transformers to predict tokens of future frames. With diffusion models thriving in the realm of image generation, the extensions of diffusion models for video prediction have been research highlights (Ho et al., 2022b;a; Xing et al., 2024; Gupta et al., 2025). Text-guided generative video prediction algorithms (Fu et al., 2023; Gu et al., 2023; Zhang et al., 2023b; Chen et al., 2025) have been designed to complete video clips under the guidance of text.

