# OpenReview forum: "Visual Representation Learning for World Models by Predicting Fine-Grained Motion"
_ICLR.cc/2025/Conference — ICLR 2025 Conference Withdrawn Submission_

### Official Review · Reviewer_KLRo · 2024-11-03

**Soundness:** 2
**Presentation:** 3
**Contribution:** 2
**Rating:** 5
**Confidence:** 3

**Summary:**

This paper tackles the problem that model-based visual RL methods tend to ignore small objects that are essential for tasks while learning irrelevant background information. The proposed approach MAWM introduces a pixel-level loss function based on motion information together with adaptive motion-aware scheduler based on timesteps and video prediction loss instead of image reconstruction as representation learning objective. Authors perform experiments on Atari and DM control benchmarks. Authors compared MAWM to several model-based and model-free RL baselines, and demonstrated improvements averaging across multiple tasks. Ablation studies show contributions of each component in this framework.

**Strengths:**

1. The proposed approach shows improvements over baselines in benchmark evaluations
2. Authors perform various ablations to demonstrate effectiveness of each component
3. Paper is well structured and experiments are well organized

**Weaknesses:**

1. The main difference from existing model-based RL methods is an auxiliary objective based on pixel-wise motion information and replacing image reconstruction loss with video prediction objective, both of are not significant changes.
2. Since the motivation of MAWM is to avoid distractions from irrelevant background information, in addition to standard control benchmarks, environments such as distracted DM control [1] and DM control generalization [2] can better demonstrate benefits of the proposed approach than baselines.
3. Current experiments do not quantitatively establish correlation whether proposed approach has more benefits in environments or tasks where objects tend to be smaller, which is a main motivation in method design.

[1] Zhang, Amy, et al. "Learning invariant representations for reinforcement learning without reconstruction." arXiv preprint arXiv:2006.10742 (2020).

[2] Hansen, Nicklas, and Xiaolong Wang. "Generalization in reinforcement learning by soft data augmentation." 2021 IEEE International Conference on Robotics and Automation (ICRA). IEEE, 2021.

**Questions:**

1. In ablation study (L486-L491), why video prediction loss instead of image reconstruction loss in MAWM framework is important for DM control experiments but not Atari experiments?
2. What is the intuition and rationale to use an adaptive Gaussian mixture model to extract ground truth label of whether each pixel belongs to foreground or background?
3. How does MAWM compare to MWM [3] that uses MAE objective on convolution features to better learn representations, which is also shown specifically more suitable for small objects?

[3] Seo, Younggyo, et al. "Masked world models for visual control." Conference on Robot Learning. PMLR, 2023.

---

> ### Author Response · Authors · 2024-11-28
> **Official Comment by Authors (1/4)**
>
> >**W1**: The main difference from existing model-based RL methods is an auxiliary objective based on pixel-wise motion information and replacing image reconstruction loss with video prediction objective, both of which are not significant changes.
>
> **R1**: Thanks for your feedback. It seems that there is some misunderstanding in our contributions. We have rewritten relevant sentences in our revised version for clarity. We hope the revised manuscript has addressed your concerns. Nevertheless, we highlight our contributions here to address your concerns:
> 1. To our best knowledge, MAWM is the first world model that incorporates a new motion-aware mechanism. Our proposed elements related to the mechanism are the motion predictor, the video predictor, AMAS, Equation 6, and Equation 8.
> 2. The adaptive motion-aware scheduler is a novel idea that imitates the dizziness mechanism of humans[1], which overcomes the shortcomings of pixel-level motion prediction when it comes to drastic changes in the environment.
> 3. As a cornerstone of world models, vanilla RSSM[2] and its variants were limited to image reconstruction[3][4][5][6]. Therefore, we propose a theoretical model, RSSM-VP, which establishes the foundation of learning the dynamics model RSSM via video prediction. Our idea can be conveniently adopted by these methods via a substitution of video prediction for image reconstruction, in which we expect significant improvement and present a viewpoint that image reconstruction is not enough for learning RL agents.
>
> **Reference**:
>
> [1]Behrang Keshavarz, Brandy Murovec, Niroshica Mohanathas, and John F Golding. The visually induced motion sickness susceptibility questionnaire (vimssq): estimating individual susceptibility to motion sickness-like symptoms when using visual devices. *Human factors*, 65(1):107–124,2023.
>
> [2]Danijar Hafner, Timothy Lillicrap, Ian Fischer, Ruben Villegas, David Ha, Honglak Lee, and James Davidson. Learning latent dynamics for planning from pixels. In *International conference on machine learning*, pp. 2555–2565. PMLR, 2019b.
>
> [3]Danijar Hafner, Jurgis Pasukonis, Jimmy Ba, and Timothy Lillicrap. Mastering diverse domains through world models. *arXiv preprint arXiv:2301.04104*, 2023.
>
> [4]Jeongsoo Ha, Kyungsoo Kim, and Yusung Kim. Dream to generalize: Zero-shot model-based reinforcement learning for unseen visual distractions. In *Proceedings of the AAAI Conference on Artificial Intelligence*, volume 37, pp. 7802–7810, 2023.
>
> [5]Christian Gumbsch, Noor Sajid, Georg Martius, and Martin V Butz. Learning hierarchical world models with adaptive temporal abstractions from discrete latent dynamics. In *International Conference on Learning Representations*, 2023.
>
> [6]Ruixiang Sun, Hongyu Zang, Xin Li, and Riashat Islam. Learning latent dynamic robust representations for world models. In *Proceedings of the 41st International Conference on Machine Learning*, Proceedings of Machine Learning Research, pp. 47234–47260. PMLR, 2024.
>
>
> >**W2**: Since the motivation of MAWM is to avoid distractions from irrelevant background information, in addition to standard control benchmarks, environments such as distracted DM control and DM control generalization can better demonstrate benefits of the proposed approach than baselines.
>
> **R2**: Thanks for your constructive advice and two additional references ([1] and [2]). We have included them in our revised version. Following your suggestion, we conduct experiments on a newer version of DM control generalization[2], DMC-GB2[3]. As described in Appendix L, the experiments demonstrate that MAWM is competitive with SADA[3], a state-of-the-art algorithm designed specifically for the benchmark, with the same settings as on the standard DeepMind Control Suite. Nevertheless, the generalization ability of MAWM on the DMC-GB2 benchmark indicates that MAWM has the potential to master a broader range of environments.
>
> **References**:
>
> [1]Amy Zhang, Rowan Thomas McAllister, Roberto Calandra, Yarin Gal, and Sergey Levine. Learning invariant representations for reinforcement learning without reconstruction. In *International Conference on Learning Representations*, 2021.
>
> [2]Nicklas Hansen and Xiaolong Wang. Generalization in reinforcement learning by soft data augmentation. In *2021 IEEE International Conference on Robotics and Automation (ICRA)*, pp.13611–13617. IEEE, 2021.
>
> [3]Abdulaziz Almuzairee, Nicklas Hansen, and Henrik I. Christensen. A recipe for unbounded data augmentation in visual reinforcement learning, *arXiv preprint arXiv:2405.17416*, 2024.

---

> ### Author Response · Authors · 2024-11-28
> **Official Comment by Authors (2/4)**
>
> >**W3**: Current experiments do not quantitatively establish correlation whether the proposed approach has more benefits in environments or tasks where objects tend to be smaller, which is a main motivation in method design.
>
> **R3**: Thank you for your feedback. To address your concern, in Appendix M, we compare MAWM with pretrained generative video on two games, Pong and Breakout, where correct prediction of moving tiny objects is essential to policy learning. Pong is a "tennis like" game that features two paddles and a tiny ball. In Breakout, the agent controls a paddle to bounce a tiny ball into bricks to destroy them. As illustrated in Figure 13 and Figure 14, MAWM can make fine-grained predictions of future frames and succeeds in forecasting the future positions of moving objects.
>
> >**Q1**: In ablation study (L486-L491), why video prediction loss instead of image reconstruction loss in MAWM framework is important for DM control experiments but not Atari experiments?
>
> **A1**: Thank you for the insightful observations. We observe that a drastic stochastic change such as a flash of light often takes place in the video of an Atari game. It is hard for world models to predict future frames under such an unnatural change. In comparison, for DM control experiments, the result of an action is deterministic. Therefore, video prediction is essential to guarantee of understanding the relationship between actions and future states.
>
> >**Q2**: What is the intuition and rationale to use an adaptive Gaussian mixture model to extract ground truth label of whether each pixel belongs to foreground or background?
>
> **A2**: We simply choose an adaptive Gaussian mixture model for its efficiency to keep consistent with our lightweight motion-aware mechanism. We did try a frame difference method but it didn't perform well. Although the adaptive gaussian mixture model is designed for scenes with a static camera, we find our proposed method works well with the ground truth label predicted by the model, even on those Atari games with a moving camera like Battle Zone. Optical flow methods may work but demand on more computational resources[1], as discussed in Appendix N.
>
> **References**:
>
> [1]Syed Tafseer Haider Shah and Xiang Xuezhi. Traditional and modern strategies for optical flow: an investigation. *SN Applied Sciences*, 3(3):289, 2021.

---

> ### Author Response · Authors · 2024-11-28
> **Official Comment by Authors (3/4)**
>
> >**Q3**: How does MAWM compare to MWM that uses MAE objective on convolution features to better learn representations, which is also shown specifically more suitable for small objects?
>
> **A3**: Thank you for your useful suggestion about the choice of the autoencoder. Following your advice, we try the MAE[1] objective on convolution features in the same way as MWM[2] does. We have included the results in Appendix F.2.
>
> Results in Table 1 demonstrate that our variational autoencoder for video ensures consistent better performance than the model that uses the MAE objective on convolution features on tasks from DeepMind Control Suite.
> Equipped with the AMAS and the motion predictor, the MAE model has an 18% performance gain, which demonstrates the effectiveness of the two key components of our proposed method. Interestingly, we find that the same MAE model performs poorly on the Atari 100k benchmark, as listed in Table 2. Our intuitive answer is that the MAE model suffers from drastic changes of images on the Atari games. After adding the AMAS and the motion predictor to the MAE model, it achieves abou 44% performance gain over the original MAE model, which demonstrates the effectiveness of the AMAS and the motion predictor again.
>
> Table 1: Ablation studies on VAE for video on eight challenging tasks from DeepMind Control Suite. AMASMO: AMAS and motion predictor.
> | Task                    | TD-MPC2           | MAE               | MAE + AMASMO      | MAWM(Ours)        |
> |:-----------------------|:-----------------:|:-----------------:|:-----------------:|:-----------------:|
> | Acrobot Swingup         | 295.3             | 236.6             | 416.1| **452.1**    |
> | Cartpole Swingup Sparse | **790.0**    | 472.9             | 548.7             | 666.7 |
> | Cheetah Run             | 537.3             | 565.7             | 765.3 | **874.3**   |
> | Finger Turn Hard        | 885.2 | 433.4             | 856.5             | **935.0**    |
> | Hopper Hop              | 302.9             | 52.5              | **399.3**    | 311.5 |
> | Quadruped Run           | 283.1             | **860.3**    | 537.0             | 648.7 |
> | Quadruped Walk          | 323.5             | **883.7**    | 835.3 | 580.3             |
> | Reacher hard            | **909.6**    |705.0 | 627.3             | 654.9             |
> | Mean($\uparrow$)        | 540.9             | 526.3             | 623.2 | **640.4**    |
> | Median($\uparrow$)      | 430.4             | 519.3             | 588.0 | **651.8**    |
>
> Table 2: Ablation studies on VAE for video on Atari 100k benchmark. AMASMO: AMAS and motion predictor.
>
> |  Game |     MAE| MAE + AMASMO  | MAWM(ours) |
> |---------------|:---------:|:---------:|:----------:|
> | Alien           | 568.5     | **952.2**     | 776.4      |
> | Amidar          | 98.5      | 117.3     | **144.2**      |
> | Assault         | 557.9     | 592.5     | **883.4**      |
> | Asterix         | 807.7     | 969.2     | **1096.9**     |
> | BankHeist       | 61.6      | 102.3     | **742.6**      |
> | BattleZone      | 6540.0    | 7543.3    | **13372.0**    |
> | Boxing          | 35.1      | **88.0**      | 85.4       |
> | Breakout        | 6.8       | 13.8      | **71.8**       |
> | ChopperCommand  | 810.0     | 79.3      | **904.0**      |
> | CrazyClimber    | 40567.0   | 44975.3   | **89038.6**    |
> | DemonAttack     | 159.8     | **313.9**     | 152.2      |
> | Freeway         | 0.0       | **0.1**       | 0.0        |
> | Frostbite       | 782.6     | **1202.7**    | 692.6      |
> | Gopher          | 633.8     | 2254.0    | **4415.8**     |
> | Hero            | 3441.1    | 6474.4    | **8801.8**     |
> | JamesBond       | 272.8     | **514.3**     | 337.2      |
> | Kangaroo        | 3577.3    | 1706.7    | **3875.6**     |
> | Krull           | 9724.1    | **10054.0**   | 8729.6     |
> | KungFuMaster    | 20902.3   | **29653.7**   | 23434.6    |
> | MsPacman        | 1092.2    | 1517.2    | **1580.7**     |
> | Pong            | 4.5       | 19.7      | **20.1**       |
> | PrivateEye      | -123.2    | **1225.5**    | -472.5     |
> | Qbert           | 912.7     | **3984.0**    | 1664.4     |
> | RoadRunner      | 7938.3    | **12548.0**   | 12518.6    |
> | Seaquest        | 635.1     | 405.3     | **557.9**      |
> | UpNDown         | 4203.0    | 3871.9    | **28408.2**    |
> |#Superhuman($\uparrow$)|5|7|**12**|
> | Mean($\uparrow$)         | 0.714     | 1.031     | **1.290**      |
> | Median($\uparrow$)       | 0.144     | 0.277     | **0.651**      |
>
> Thank you so much for your insightful and constructive suggestions about these additional experiments, which provide strong evidence for the effectiveness of our three key components and the generalization ability of MAWM.

---

> ### Author Response · Authors · 2024-12-02
> **Official Comment by Authors (4/4)**
>
> **References**:
>
> [1]Kaiming He, Xinlei Chen, Saining Xie, Yanghao Li, Piotr Dollar, and Ross Girshick. Masked autoencoders are scalable vision learners. In *Proceedings of the IEEE/CVF conference on computer vision and pattern recognition*, pp. 16000–16009, 2022.
>
> [2]Younggyo Seo, Danijar Hafner, Hao Liu, Fangchen Liu, Stephen James, Kimin Lee, and Pieter Abbeel. Masked world models for visual control. In *Conference on Robot Learning*, pp. 1332–1344. PMLR, 2023.
>
> ---
> We hope we have addressed your concerns regarding our contributions, the relationship between fine-grained prediction and performance, and the additional benchmark.

---

### Official Review · Reviewer_pZ6p · 2024-11-03

**Soundness:** 2
**Presentation:** 2
**Contribution:** 2
**Rating:** 5
**Confidence:** 3

**Summary:**

This paper introduces a new method for learning the model in model-based RL with special considerations in motion prediction modeling. The authors introduces motion awareness by adding foreground motion prediction and adaptive motion-blur losses over traditional video generative pipelines. The resulting model achieves comparable results against existing MBRL methods.

**Strengths:**

The proposed motion awareness in model learning is intuitive and the proposed method achieves comparative results with state-of-the-art methods.

**Weaknesses:**

One concern about this paper is the significance of the proposed method. First, except in curves and Fig.3 from the ablation studies, the authors might want to provide more visualizations on the effect of the motion awareness introduced, especially when considering the current experiment setting is only in atari and dm-control-suite where data are all synthetic and should potentially be simpler compared to real-world videos. Second, I wonder if the same pipeline could be applied to real-world videos as currently there is increasing trend in leveraging video generative models as "world models" to facilitate various tasks. It would be better if the authors could find a proper way to compare or address such line of methods. Lastly, the paper organization could be improved for better clarity.

**Questions:**

See the weakness section.

---

> ### Author Response · Authors · 2024-11-28
> **Official Comment by Authors (1/2)**
>
> We thank the reviewer for the detailed feedback.
>
> >**Q1**: One concern about this paper is the significance of the proposed method.
>
> **A1**: Our proposed method is of significance:
> 1. We propose a novel idea about the motion-aware mechanism and incorporate it into our world model, MAWM, which learns compact visual representations via motion prediction and an Adaptive Motion-Aware Scheduler (AMAS). To our best knowledge, no motion-aware mechanism has ever been applied to world models and our proposed motion-aware mechanism can be incorporated into existing MBRL methods to capture moving tiny objects, which we believe could be a source of inspiration for other MBRL methods.
> 2. As a significant cornerstone of world models, vanilla RSSM[1] and its variants were limited to image reconstruction[2][3][4][5]. Therefore, we propose a theoretical model, RSSM-VP, which establishes the foundation of learning RSSM the dynamics model via video prediction. Our idea can be conveniently adopted by these methods via the substitution of video prediction for image reconstruction, in which we expect significant improvement.
> 3. We evaluate MAWM on 46 tasks (including all the 20 tasks on DeepMind Control Suite in our revised version) across diverse domains with fixed hyperparameters and demonstrate its consistent significant improvement over baselines. Significantly, MAWM outperforms TD-MPC2, the state-of-the-art RL algorithms without a lookahead search on the DeepMind Control Suite, by a large margin. Furthermore, we further demonstrate the generalization ability of MAWM in DMC-GB2, where test environments are visually distinct from the training environment.
>
> **References**:
>
> [1]Danijar Hafner, Timothy Lillicrap, Ian Fischer, Ruben Villegas, David Ha, Honglak Lee, and James Davidson. Learning latent dynamics for planning from pixels. In *International conference on machine learning*, pp. 2555–2565. PMLR, 2019b.
>
> [2]Danijar Hafner, Jurgis Pasukonis, Jimmy Ba, and Timothy Lillicrap. Mastering diverse domains through world models. *arXiv preprint arXiv:2301.04104*, 2023.
>
> [3]Jeongsoo Ha, Kyungsoo Kim, and Yusung Kim. Dream to generalize: Zero-shot model-based reinforcement learning for unseen visual distractions. In *Proceedings of the AAAI Conference on Artificial Intelligence*, volume 37, pp. 7802–7810, 2023.
>
> [4]Christian Gumbsch, Noor Sajid, Georg Martius, and Martin V Butz. Learning hierarchical world models with adaptive temporal abstractions from discrete latent dynamics. In *International Conference on Learning Representations*, 2023.
>
> [5]Ruixiang Sun, Hongyu Zang, Xin Li, and Riashat Islam. Learning latent dynamic robust representations for world models. In *Proceedings of the 41st International Conference on Machine Learning*, Proceedings of Machine Learning Research, pp. 47234–47260. PMLR, 2024.
> >**Q2**: First, except in curves and Fig.3 from the ablation studies, the authors might want to provide more visualizations on the effect of the motion awareness introduced, especially when considering the current experiment setting is only in atari and dm-control-suite where data are all synthetic and should potentially be simpler compared to real-world videos.
>
> **A2**: Thanks for your suggestions. We provide Appendix M in the revised version, for visualizations of video prediction by MAWM, comparing with pretrained video generation models. Results show that the MAWM can capture the moving patterns of tiny objects. However, pretrained Stable Diffusion model fails in these cases. Furthermore, as showcased in Appendix L, experiments on DMC-GB2, where test environments are visually distinct from the training environment and real-world video serve as the background of the environment, demonstrate the generalization ability of MAWM, and indicate its potential to work in a real-world application.

---

> ### Author Response · Authors · 2024-11-28
> **Official Comment by Authors (2/2)**
>
> >**Q3**: I wonder if the same pipeline could be applied to real-world videos as currently there is increasing trend in leveraging video generative models as "world models" to facilitate various tasks. It would be better if the authors could find a proper way to compare or address such line of methods.
>
> **A3**: This is an interesting suggestion. Currently, there is a gap between world models in the realm of MBRL and "World Models" for video generative models like Sora in that ontology of relationship among images, actions and rewards is different. A world model in MBRL settings is a generative model that produces future states and rewards, *i.e.*, models of $p(s_{t+1}, r_{t} | s_t, a_t)$[1]. Diamond[2] trains a diffusion model and a reward model separately for RL agents, the training pipeline of which is the same as "World Models" for video generative models de facto. It is not trivial to combine reward prediction and future state generation with extracted representations from video generative models[3] for future work. Nevertheless, computation resources should be taken into consideration for academic research. Please refer to Appendix I for a comparison of computation resources. Under comparative computation time with DIAMOND on a NVIDIA GeForce RTX 4090, we have obtained results on the following ten games up till now:
> | Game |DIAMOND(100k)|MAWM(580k)|
> | --- | --- | --- |
> | Boxing |86.9 |**95.0**|
> | Breakout |  132.5 |**414** |
> |CrazyClimber| 99167.8 |**114529.0** |
> | DemonAttack | 288.1|**2225.3** |
> |Frostbite|274.1 |**3507.1** |
> | Gopher|5897.9|**25104.2** |
> |KungFuMaster|23523|**28730.0**|
> |PrivateEye|114.3 |**5412.1** |
> |Seaquest|551.2 |**1381.4** |
> |MsPacman|1958.2|**2699.5**|
>
> We will run more experiments in the near future if you are interested in our results.
>
> **References**:
>
> [1]David Ha and Jurgen Schmidhuber. Recurrent world models facilitate policy evolution. Advances in neural information processing systems, 31, 2018.
>
> [2]Eloi Alonso, Adam Jelley, Vincent Micheli, Anssi Kanervisto, Amos Storkey, Tim Pearce, and Franc¸ois Fleuret. Diffusion for world modeling: Visual details matter in atari. In Thirty-eighth Conference on Neural Information Processing Systems, 2024.
>
> [3]Grace Luo, Lisa Dunlap, Dong Huk Park, Aleksander Holynski, and Trevor Darrell. Diffusion hyperfeatures: Searching through time and space for semantic correspondence. Advances in Neural Information Processing Systems, 36, 2024.
> >**Q4**: Lastly, the paper organization could be improved for better clarity.
>
> **A4**: Thanks for your suggestion. We highlight important modifications in blue and reorganize the appendices in our revised version.
>
> ---
> We hope to have addressed your concerns regarding the significance of our method and the comparison with the other line of methods.

---

### Official Review · Reviewer_qY1E · 2024-11-04

**Soundness:** 3
**Presentation:** 3
**Contribution:** 2
**Rating:** 5
**Confidence:** 3

**Summary:**

The paper presents a novel approach in terms of model-based reinforcement learning (MBRL), where an image-based world model is utilized to improve sample efficiency by mimicking a scalable digital copy of the environment, such as DreamerV3. The authors claims that traditional world models tend to ignore moving tiny objects and their connections with tasks, thereby they propose a Motion-Aware World Model (MBRL) to account for them. Specifically, MAWM 1. focuses on small objects via pixel-level attention mechanisms and 2. deals with rapid changes of them via an adaptive control scheduler. The results on Atari 100k and DeepMind Control Suite (DMC) depict the superiority of the proposed method against DreamV3.

**Strengths:**

The paper presents several strengths:
1. The experimental validation of the proposed model on two standard datasets, Atari 100k and DMC, demonstrates its effectiveness and generalizability.
2. The writing of the paper is clear, making it easy for readers to understand the proposed method and its experimental validation.
3. The details of the model in the appendix are valuable contributions to the community, as it enables other researchers to reproduce the results and build upon the proposed method.

**Weaknesses:**

1. One concern is regarding the novelty of the proposed two techniques. The motion-aware auxiliary loss is not a novel topic [1]. Additionally, the technical contributions of the proposed pixel-level attention and scheduler are not clear enough to me.
[1] 3D Motion Decomposition for RGBD Future Dynamic Scene Synthesis, CVPR 2019.

2. Is it possible to compare MAWM with pretrained video generation models like stable video diffusion, to figure out whether they can capture the patterns of moving targets or not?

3. I also have some concerns about the experimental results. As an example, Table 1 is confusing. In the last row, the Median score of a counterpart SimPLe is 1.34, which seems to obtain the best results without being marked bold. Besides, the different components proposed in this paper don’t demonstrate convincing improvement in Figure 4. A more comprehensive and significant ablation study could help address this.

**Questions:**

Please see the weakness

---

> ### Author Response · Authors · 2024-11-28
> **Official Comment by Authors (1/2)**
>
> We thank the reviewer for the detailed feedback.
>
> ---
> >**Q1**: One concern is regarding the novelty of the proposed two techniques. The motion-aware auxiliary loss is not a novel topic [1]. Additionally, the technical contributions of the proposed pixel-level attention and scheduler are not clear enough to me.
>
>
> **A1**: Thank you for the additional references[1]. We have included it in the revised version. We have updated Appendix N to include related works according to your advice. As discussed below, we think our motion-aware visual representation learning in MAWM is orthogonal to the mentioned method.
> 1. The purpose of introducing a motion-aware mechanism is different. We concern about learning compact and meaningful representations for policy learning within limited interactions with the environment, while the mentioned method studies RGBD future scene synthesis.
> 2. The way we predict motion is distinct from the mentioned method. We integrate into world models the lightweight motion decoder to predict fine-grained future motion and use an adaptive GMM model to generate the "ground truth". In comparison, the paper uses point clouds from the last two depth maps to calculate the current change in camera pose and foreground motion. Via predicted future changes in camera pose and foreground motion by two relevant neural networks, the paper calculates 3D point clouds in the next frame, which represents the future location of pixels.
> 3. The training objective of our motion-aware mechanism is different from the proposed method. We explicitly minimize the focal loss for motion prediction. However, the paper implicitly learns to predict future motion via the estimation of images, semantic maps, and depth maps, which are further input into a refinement network to generate refined results.
>
> We admit that developing a motion-aware mechanism is an existing research topic in the field of computer vision. However, developing a universal appropriate auxiliary loss is a challenging problem in the field of MBRL[2]. Since the ground truth for motion clues cannot be computed or obtained in advance under MBRL settings, it is essential to develop a lightweight and efficient motion-aware mechanism. In conclusion, our technical contributions are threefold:
> 1. To our best knowledge, MAWM is the first world model that incorporates a new motion-aware mechanism, which we believe could be a source of inspiration for other MBRL methods.
> 2. Nevertheless, the adaptive motion-aware scheduler is a novel idea that imitates the dizziness mechanism of humans, which overcomes the shortcomings of pixel-level motion prediction when it comes to drastic changes in the environment.
> 3. Moreover, all MBRL methods that use RSSM[3] as the dynamics model train world models via image reconstruction due to a lack of theoretical support for video prediction. Thus, we propose the theoretical model RSSM-VP and demonstrate its efficiency and performance over the vanilla RSSM in ablation studies.
>
> **References**:
>
> [1] Xiaojuan Qi, Zhengzhe Liu, Qifeng Chen, and Jiaya Jia. 3d motion decomposition for rgbd future dynamic scene synthesis. In *Proceedings of the IEEE/CVF Conference on Computer Vision and Pattern Recognition*, pp. 7673–7682, 2019.
>
> [2] Thomas M Moerland, Joost Broekens, Aske Plaat, Catholijn M Jonker, et al. Model-based reinforcement learning: A survey. *Foundations and Trends® in Machine Learning*, 16(1):1–118, 2023.
>
> [3] Danijar Hafner, Timothy Lillicrap, Ian Fischer, Ruben Villegas, David Ha, Honglak Lee, and James Davidson. Learning latent dynamics for planning from pixels. In *International conference on machine learning*, pp. 2555–2565. PMLR, 2019b.
>
> ---
> >**Q2**: Is it possible to compare MAWM with pretrained video generation models like stable video diffusion, to figure out whether they can capture the patterns of moving targets or not?
>
> **A2**: Thanks for your useful suggestion. We list the best results of pretrained video generation models in Figure 13 and Figure 14. Results show that they are often incapable of catching the patterns of moving targets, while MAWM can make fine-grained future frame predictions.

---

> ### Author Response · Authors · 2024-11-28
> **Official Comment by Authors (2/2)**
>
> >**Q3**: I also have some concerns about the experimental results. As an example, Table 1 is confusing. In the last row, the Median score of a counterpart SimPLe is 1.34, which seems to obtain the best results without being marked bold. Besides, the different components proposed in this paper don’t demonstrate convincing improvement in Figure 4. A more comprehensive and significant ablation study could help address this.
>
> **A3**: Thank you for pointing out the mistake. The Median score of SimPLe should be $0.134$, and we have corrected it in the revised version. For ablation studies, we randomly selected 6 tasks for Atari 100k and 4 tasks for DeepMind Control Suite with fixed hyperparameters across both domains. We have revised the description for Figure 4 according to your advice. We add additional comprehensive ablation studies on the effect of relevant modules, which can be found in Appendix F. Furthermore, we run experiments on all the 20 tasks on DeepMind Control Suite for a more comprehensive study of MAWM.
>
> ---
> We hope we have addressed your concerns about the novelty of our method and the comparison of prediction results with pretrained models.

---

### Official Review · Reviewer_vTbC · 2024-11-04

**Soundness:** 3
**Presentation:** 2
**Contribution:** 2
**Rating:** 5
**Confidence:** 3

**Summary:**

This paper presents the Motion-Aware World Model. It integrates a fine-grained motion predictor with an adaptive motion-aware scheduler and involves action-conditional video prediction to filter out backgrounds and track object motion at the pixel level.

**Strengths:**

1. The attention that the authors give to relevant foreground information and moving tiny objects is meaningful for constructing world models.

2. The performance gain of the proposed method is considerable on the Atari 100K benchmark and DeepMind Control Suite.

**Weaknesses:**

1. The author elaborates on some details in the method section; however, many technical aspects, like the Convolutional Block Attention Module, were not employed in previous approaches. Nevertheless, the relevant experiments have not undergone ablation, which could readily prompt people to doubt the fairness of the experiments.

2. The results of the ablation experiments are rather confusing. It appears that MAWM is not consistently the best choice. Moreover, for the majority of the experiments, it doesn't seem that they have reached convergence.

3. The motivation of this paper is to solve the problem of struggling with irrelevant background information and omitting moving tiny objects. Is there any corresponding visualization to verify this?

4. Many experimental details are not clearly stated. For example, what are the values of $\alpha$ and $r_{dizzy}$?

5. Discussions on limitations are necessary.

**Questions:**

Please see weaknesses above.

---

> ### Author Response · Authors · 2024-11-28
> **Official Comment by Authors (1/3)**
>
> We thank the reviewer for the detailed feedback.
>
> ---
>
> >**Q1**: The author elaborates on some details in the method section; however, many technical aspects, like the Convolutional Block Attention Module, were not employed in previous approaches. Nevertheless, the relevant experiments have not undergone ablation, which could readily prompt people to doubt the fairness of the experiments.
>
> **A1**: We agree that relevant modules need ablation studies to justify our contributions compared with previous works in MBRL. We have included an ablation study on the two modules (*i.e.*, convolutional block attention module and harmonizer) in Appendix F.1. Without these modules, MAWM still outperforms the best baselines on both the benchmarks, as shown in Table 1 and Table 2, which demonstrate the effectiveness of our proposed method.
>
> Table 1: Ablation studies on CBAM and Harmonizers on the Atari 100k benchmark. Both: CBAM
> and Harmonizers, Standard: standard configurations of MAWM in the body of our paper.
> |Game | REM | - Both    | - CBAM    | MAWM(ours) |
> |-----------------------|---------|----------|----------|-----------|
> | Alien           | 607.2    | 1089.0    | **1165.4**    | 776.4      |
> | Amidar          | 95.3     | **210.9**     | 110.8     | 144.2      |
> | Assault         | **1764.2**   | 1075.1    | 790.9     | 883.4      |
> | Asterix         | **1637.5**   | 1466.3    | 1201.8    | 1096.9     |
> | BankHeist       | 19.2     | 517.2     | **987.5**     | 742.6      |
> | BattleZone      | 11826    | 8060.0    | 10696.7   | **13372.0**    |
> | Boxing          | **87.5**     | 80.9      | 84.2      | 85.4       |
> | Breakout        | 90.7     | **108.7**     | 40.6      | 71.8       |
> | ChopperCommand  | **2561.2**   | 899.0     | 818.0     | 904.0      |
> | CrazyClimber    | 76547.6  | 82506.7   | **89538.3**   | 89038.6    |
> | DemonAttack     | **5738.6**   | 149.1     | 157.4     | 152.2      |
> | Freeway         | **32.3**     | 0.0       | 0.0       | 0.0        |
> | Frostbite       | 240.5    | 2040.0    | **2449.2**    | 692.6      |
> | Gopher          | 5452.4   | 3403.1    | **8012.3**    | 4415.8     |
> | Hero            | 6484.8   | **11482.4**   | 8139.8    | 8801.8     |
> | JamesBond       | 391.2    | **477.0**     | 376.3     | 337.2      |
> | Kangaroo        | 467.6    | 1726.7    | 1836.0    | **3875.6**     |
> | Krull           | 4017.7   | 8312.8    | 8408.5    | **8729.6**     |
> | KungFuMaster    | **25172.2**  | 19122.7   | 21415.3   | 23434.6    |
> | MsPacman        | 962.5    | 1557.3    | 1573.7    | **1580.7**     |
> | Pong            | 18       | **20.2**      | 18.3      | 20.1       |
> | PrivateEye      | 99.6     | **3288.6**    | 1423.8    | -472.5     |
> | Qbert           | 743      | **4237.2**    | 1145.1    | 1664.4     |
> | RoadRunner      | 14060.2  | **20635.7**   | 14725.3   | 12518.6    |
> | Seaquest        | **1036.7**   | 440.0     | 554.0     | 557.9      |
> | UpNDown    | 3757.6   | 15716.1   | 15952.4   | **28408.2**    |
> | #Superhuman($\uparrow$)  |  **12** | 10     |11     | **12**  |
> | Mean($\uparrow$) | 1.222    | 1.289     | 1.258     | **1.290**      |
> | Median($\uparrow$) | 0.280    | 0.512     | 0.578     | **0.651**      |
>
> Table 2: Ablation studies on CBAM and Harmonizers on eight challenging tasks from DeepMind Control
> Suite. Both: CBAM and Harmonizers, Standard: standard configurations of MAWM in the body of our paper.
> | Task                     | TD-MPC2 | - Both  | - CBAM  | MAWM(ours) |
> |------------------------|:-------:|:-------:|:-------:|:----------:|
> | Acrobot Swingup          | 295.3   | 412.3   | 427.0   | **452.1**     |
> | Cartpole Swingup Sparse  | **790.0**  | 519.2   | 603.9   | 666.7      |
> | Cheetah Run              | 537.3   | 899.3   | **915.9**   | 874.3      |
> | Finger Turn Hard         | 885.2   | **935.9**  | 825.2   | 935.0      |
> | Hopper Hop               | 302.9   | 334.8   | **355.2**   | 311.5      |
> | Quadruped Run            | 283.1   | 577.9   | 644.1   | **648.7**     |
> | Quadruped Walk           | 323.5   | 620.7   | **653.2**   | 580.3      |
> | Reacher Hard             | **909.6**   | 582.8   | 689.3   | 654.9      |
> |  Mean($\uparrow$) | 540.9   | 610.4   | 639.2   | **640.4**      |
> | Median($\uparrow$)| 430.4   | 580.4   | 648.6   | **651.8**      |

---

> ### Author Response · Authors · 2024-11-28
> **Official Comment by Authors (2/3)**
>
> >**Q2**: The results of the ablation experiments are rather confusing. It appears that MAWM is not consistently the best choice. Moreover, for the majority of the experiments, it doesn't seem that they have reached convergence.
>
> **A2**: It is a common setting for sample efficiency in related work, such as IRIS[1], DreamerV3[2] and EfficientZero V2[3]. On the Atari 100k benchmark, the agent is limited to fixed 100k interactions with the environment for each task. For comparison, unconstrained agents on Atari Games often require a budget of 50M interactions with environments while the computational resources of academic institutions are often limited. Nevertheless, we agree that training sample-efficient methods with a more computational budget is essential to guarantee that performance will improve consistently with more samples. To that end, we conducted an additional experiment with 1M training steps on Breakout, Demon Attack, Gopher. The game scores and human-normalized scores for the two games are listed below:
> | Game |Score(100k)|HNS(100k)|Score(1M)|HNS(1M)|
> | --- | --- | --- | --- | --- |
> | Breakout | 71.8 | 2.435 |  422.4 |14.608 |
> | DemonAttack | 152.2 | 0.000 | 2374.8| 1.222|
> | Gopher | 4415.8 | 1.930 | 99995.4| 46.284 |
>
> Just with more samples, MAWM at 1M steps achieves 6 times the game score of Breakout, 16 times the game score of Demon Attack, and 23 times the game score of Gopher at 100k steps. We will run experiments on all 26 Atari games for the final revision.
>
> To save the hassle of tuning hyperparameters for each task, there is a growing trend in the development of MBRL algorithms that use fixed hyperparameters across all tasks[4]. Thus, no method can always achieve the best performance in every task for now, as it is shown in Table 11. For the fairness of the ablation studies, we randomly selected 6 tasks for Atari 100k and 4 tasks for DeepMind Control Suite. Results show that MAWM is the best choice for the majority of experiments. Nevertheless, it is interesting to develop an algorithm with fixed hyperparameters that works satisfactorily for all tasks in future work.
>
> **References**:
>
> [1]Vincent Micheli, Eloi Alonso, and Franc¸ois Fleuret. Transformers are sample-efficient world models. In *International Conference on Learning Representations*, 2023.
>
> [2]Danijar Hafner, Jurgis Pasukonis, Jimmy Ba, and Timothy Lillicrap. Mastering diverse domains through world models. *arXiv preprint arXiv:2301.04104*, 2023.
>
> [3]Shengjie Wang, Shaohuai Liu, Weirui Ye, Jiacheng You, and Yang Gao.
> EfficientZero v2: Mastering discrete and continuous control with limited data. In *Proceedings of the 41st International Conference on Machine Learning*, volume 235 of Proceedings of Machine Learning Research, pp. 51041–51062. PMLR, 21–27 Jul 2024.
>
> [4]Nicklas Hansen, Hao Su, and Xiaolong Wang. Td-mpc2: Scalable, robust world models for continuous control. In *International Conference on Learning Representations*, 2024.
>
> >**Q3**: The motivation of this paper is to solve the problem of struggling with irrelevant background information and omitting moving tiny objects. Is there any corresponding visualization to verify this?
>
> **A3**: Thanks for your valuable suggestion. We have included Figure 13 and Figure 14 in the revised manuscript. The visualization shows the MAWM can capture the moving patterns of tiny objects. However, the pretrained Stable Diffusion model fails in these cases.
>
> >**Q4**: Many experimental details are not clearly stated. For example, what are the values of $\alpha$ and $r_\text{dizzy}$?
>
> **A4**: We listed all the experimental details in appendices in the previously submitted manuscript. For example, as listed in Table 8, $\alpha=0.15$ and $r_\text{dizzy}=0.05$, since we observe that humans focus on a relatively small area during learning. Although the hyperparameters may not be the best choice due to our limited computational resources, we do believe that our proposed motion-aware mechanism is general and can be a cornerstone for world models in future. Please refer to Appendix B for MAWM architecture, Appendix D for our hyperparameters, and Appendix I for computational resources. If you have interest in more details, don't hesitate to let us know.

---

> > ### Author Response · Authors · 2024-12-03
> > **Official Comment by Authors (3/3)**
> >
> > >**Q5**: Discussions on limitations are necessary.
> >
> > **A5**: We agree that it is necessary to discuss the potential limitations of our work. We have included the following discussions in the Conclusion section of our revised version. We identify three potential limitations of our work for future research:
> > 1. MAWM has difficulties in long-horizon video prediction, which is also the key problem in current MBRL methods[1]. Specifically, if the imagination step is large, predicted images may be incorrect in certain cases, even though predicted motion by MAWM remains accurate. Future work can try to find whether perfect long-horizon video prediction improves policy learning.
> > 2. Besides, although MAWM has been trained with fixed hyperparameters across domains, we currently train a standalone model for each task. An exciting avenue is to explore the potential of MAWM to finish different tasks within a model by effectively sharing common knowledge.
> > 3. Since MAWM learns task-specific relationships between actions and images, another promising avenue might be to integrate text-guided video generative models[2][3][4][5][6][7].
> >
> > **References**:
> >
> > [1]Eloi Alonso, Adam Jelley, Vincent Micheli, Anssi Kanervisto, Amos Storkey, Tim Pearce, and Franc¸ois Fleuret. Diffusion for world modeling: Visual details matter in atari. In *Thirty-eighth Conference on Neural Information Processing Systems*, 2024.
> >
> > [2]Robin Rombach, Andreas Blattmann, Dominik Lorenz, Patrick Esser, and Bjorn Ommer. High-resolution image synthesis with latent diffusion models. In *Proceedings of the IEEE/CVF conference on computer vision and pattern recognition*, pp. 10684–10695, 2022.
> >
> > [3]Tim Brooks, Aleksander Holynski, and Alexei A. Efros. Instructpix2pix: Learning to follow image editing instructions. In *Proceedings of the IEEE/CVF Conference on Computer Vision and Pattern Recognition*, pp. 18392–18402, 2023.
> >
> > [4]Lvmin Zhang, Anyi Rao, and Maneesh Agrawala. Adding conditional control to text-to-image diffusion models. In *Proceedings of the IEEE/CVF International Conference on Computer Vision*, pp. 3836–3847, 2023a.
> >
> > [5]Andreas Blattmann, Tim Dockhorn, Sumith Kulal, Daniel Mendelevitch, Maciej Kilian, Dominik Lorenz, Yam Levi, Zion English, Vikram Voleti, Adam Letts, et al. Stable video diffusion: Scaling latent video diffusion models to large datasets. *arXiv preprint arXiv:2311.15127*, 2023.
> >
> > [6]Hyeonho Jeong, Geon Yeong Park, and Jong Chul Ye. Vmc: Video motion customization using temporal attention adaption for text-to-video diffusion models. In *Proceedings of the IEEE/CVF Conference on Computer Vision and Pattern Recognition*, pp. 9212–9221, 2024.
> >
> > [7]Grace Luo, Lisa Dunlap, Dong Huk Park, Aleksander Holynski, and Trevor Darrell. Diffusion hyperfeatures: Searching through time and space for semantic correspondence. *Advances in Neural Information Processing Systems*, 36, 2024.
> >
> > ---
> > We hope we have addressed your concerns about ablation studies and whether our model improves with more interactions with environments.

---

### Author Response · Authors · 2024-11-30
**General Response**

We thank the reviewers for their detailed reviews. Based on their comments, we have revised our paper as listed below:
- We have run experiments on an additional benchmark to demonstrate the generalization ability and benefits of our proposed approach (in Appendix L), following suggestions of Reviewer [KLRo](https://openreview.net/forum?id=8BJl6LQgW5&noteId=LKr56goOPB).
- We have added several ablation studies on CBAM, harmonizers, and choice of the autoencoder to demonstrate the effectiveness of our motion-aware mechanism and the importance of video prediction (in Appendix F), according to Reviewers [vTbC](https://openreview.net/forum?id=8BJl6LQgW5&noteId=snUvzjSHNH), [qY1E](https://openreview.net/forum?id=8BJl6LQgW5&noteId=sqISwzHENv), and [KLRo](https://openreview.net/forum?id=8BJl6LQgW5&noteId=LKr56goOPB).
- We have illustrated that our proposed methods can capture the motion of tiny objects and make almost perfect predictions, in comparison to pretrained video generation models, revealing that our motion-aware mechanism is the key to efficient performance for most tasks (in Appendix M), for Reviewers [vTbC](https://openreview.net/forum?id=8BJl6LQgW5&noteId=snUvzjSHNH), [qY1E](https://openreview.net/forum?id=8BJl6LQgW5&noteId=sqISwzHENv), [pZ6p](https://openreview.net/forum?id=8BJl6LQgW5&noteId=4dLWWd9qsd), and [KLRo](https://openreview.net/forum?id=8BJl6LQgW5&noteId=LKr56goOPB) concerns.
- We have discussed differences between our motion-aware mechanism and existing methods in the response to Reviewers [qY1E](https://openreview.net/forum?id=8BJl6LQgW5&noteId=sqISwzHENv), and extended our related work (in Appendix N).
- We have conducted an experiment to ensure that MAWM improves beyond standard settings of 100k interactions on Atari Games, implying that MAWM may have better efficiency and effectiveness than existing methods, in answer to Reviewers [vTbC](https://openreview.net/forum?id=8BJl6LQgW5&noteId=snUvzjSHNH) and [pZ6p](https://openreview.net/forum?id=8BJl6LQgW5&noteId=4dLWWd9qsd).
- We have included a discussion of the limitations of our work (in Conclusion), according to Reviewer [vTbC](https://openreview.net/forum?id=8BJl6LQgW5&noteId=snUvzjSHNH).
- We have extended experiments to show MAWM consistently satisfactory performance on all the 20 tasks from DeepMind Control Suite, setting a state-of-the-art on the whole benchmark, as showcased in Table 12, which can be a reply to [vTbC](https://openreview.net/forum?id=8BJl6LQgW5&noteId=snUvzjSHNH) and [qY1E](https://openreview.net/forum?id=8BJl6LQgW5&noteId=sqISwzHENv).
- We have reflected feedbacks on revisions of our manuscript from Reviewers [vTbC](https://openreview.net/forum?id=8BJl6LQgW5&noteId=snUvzjSHNH), [qY1E](https://openreview.net/forum?id=8BJl6LQgW5&noteId=sqISwzHENv), [pZ6p](https://openreview.net/forum?id=8BJl6LQgW5&noteId=4dLWWd9qsd), and [KLRo](https://openreview.net/forum?id=8BJl6LQgW5&noteId=LKr56goOPB).
- We showcase visualization [here](https://anonymous.4open.science/r/mawm-C555)  to help understand our intuition and rationale for our motion-aware mechanism. We will release our code to promote future work.

---

### Note · Authors · 2025-04-07

I have read and agree with the venue's withdrawal policy on behalf of myself and my co-authors.

---

### Meta-Review · Area_Chair_9eEA · 2024-12-25

**Metareview:**

The submission addresses the problem of model-based reinforcement learning in visual environments. It aims to introduce a "world model" that is better at capturing the visual dynamics of tiny objects that are relevant to the target tasks. This is achieved by adding foreground motion prediction and adaptive motion-blur losses. Evaluations are performed on Atari and Deepmind Control Suite. The submission received four borderline reject (5) ratings initially, and the authors provided their rebuttal. Unfortunately, none of the reviewers engage in the post-rebuttal discussion. After reading through the rebuttals and the submission, the AC believes that the submission should be revised to better explain its contributions with respect to prior work, along with the generalizability and limitations of the approach when applied to visually more complex scenarios. Overall, the AC finds no ground to overturn the consensus among all four reviewers.

**Additional Comments On Reviewer Discussion:**

All reviewers rated the submission as borderline reject (5) before the rebuttal, and none of them engaged in the discussion. Nonetheless, the AC shares their concerns that the contributions of the submission should be better positioned with respect to both literature on model-based RL for visual environments, as well as "video" generative modeling in the computer vision community. Additionally, the AC has reservations on the generalizability of the proposed approach towards visually more complex environments, along with environments where a diverse range of tasks could be potentially accompolished.

---

### Decision · Program_Chairs · 2025-01-22

Reject